# Ultrablack wool textiles inspired by hierarchical avian structure

Hansadi Jayamaha [1,2], Kyuin Park [1,2] & Larissa M. Shepherd [1] ✉

Wildlife often uses a combination of colors in their skin, scales, and feathers to both attract mates and avoid predators. Some animals express an extreme level of black color, called "ultrablack" with reflectance <0.5%. The various nano/microstructures that produce ultrablack have been studied and replicated synthetically. These synthetic ultrablack materials, however, use highly advanced and costly techniques, toxic substances, and lack the flexibility and biocompatibility that are often desired in real-world textile applications. Here we show that a conventional natural fabric can be transformed into an ultrablack one with a biocompatible dye and a surface modification to create nanofibrils. The ultrablack wool fabrics we report have an average total reflectance of 0.13% ($\lambda = 400–700$ nm) and represent the darkest fabrics currently reported. Unlike commercially available ultrablack fabrics, the ultrablack wool developed in this study remains breathable and conformable. Furthermore, it exhibits wide-angle ultrablack performance, maintaining a symmetric (angle-independent) optical response across a 120° angular span. Environmental and mechanical tests also prove the material's resilience, showing the retention of its natural fabric characteristics alongside its ultrablack properties. To demonstrate the manufacturability in the textile industry, we present multiple parameter sets for plasma etching to achieve the ultrablack effect.

An ultrablack material has low reflectance (<0.5%) across the visible spectrum ($\lambda = 400–700$ nm)[1]. Several species of birds[2,3], fish[1], and insects[4,5] have been discovered to have directional reflectance, normal to the surface, as low as 0.05%. These biological examples have inspired researchers to replicate nature's mechanisms[6–9]. Complex nano/microstructures reduce backscatter and total reflectance by increasing the forward scattering, optical path length, and, consequently, the number of opportunities for light absorption[5]. Our research is inspired by the plumages of birds-of-paradise with their hierarchical barbule microstructures and melanin-based internal nanostructures, exhibiting directional (0°, normal) reflectance and total reflectance as low as 0.05% and 3.14% in the visible spectrum, respectively[2]. One of the darkest man-made materials is vertically aligned nanotube array (VANTA or

VACNT) structures that reflect as low as 0.005% of visible light[10,11]. Producing these materials is a costly, highly technical process that uses hazardous substances, and the finished products are both fragile and toxic[12,13]. These ultrablack materials are unsuitable as wearable or everyday textiles that require biocompatibility, breathability, and stretchability.

Here, we demonstrate a simple and industrially viable two-step method to fabricate the darkest ultrablack fabric ever reported, which also retains the original integrity of the textile material. Our two-step method includes one-pot polydopamine (PDA) dyeing of merino wool followed by plasma treatment. This study demonstrates the fabrication and validates the properties of ultrablack wool (UBW) through microscopic and spectrophotometric analyses, surface characterization, and optical simulations. We report wash-fastness, lightfastness,

[1]Department of Human Centered Design, Cornell University, Ithaca, NY, USA. [2]These authors contributed equally: Hansadi Jayamaha, Kyuin Park.
✉e-mail: larissa.shepherd@cornell.edu

and mechanical durability, which confirm the real-world pertinency of UBW as a flexible, breathable, and wearable optical material.

## Results and discussion

### Visual and microscopic surface analyses of nano/microstructures

The magnificent riflebird (*Ptiloris magnificus*), a species of the bird of paradise family (Fig. 1a inset), has hierarchical barbules that curve up to form a densely packed tilted array (~30°) toward the distal tip of the feather (Fig. 1a)[2]. The presence of the intra-barbule grooves (~5–30 μm), with smaller cavities (<5 μm) along the barbule margins

and their melanin-based internal nanostructures, results in the ultrablack coloration that is meant for display, rather than camouflage[2]. Inspired by this structure-induced darkness, the centerpiece of the dress in Fig. 1b portrays the contrast between the ultrablack wool (UBW) against the iridescent blue and commercially available black polyester blend (BPETB) fabrics.

Figure 1c illustrates the two-step method we use to turn white merino wool (WMW) into polydopamine (PDA)-dyed merino wool (PDAMW), and finally to UBW. Unlike most man-made ultrablack materials[10,11,14–16], the merino wool, harvested from the *Ovis aries* sheep, is economical, bio-manufactured, and biocompatible. To impart black

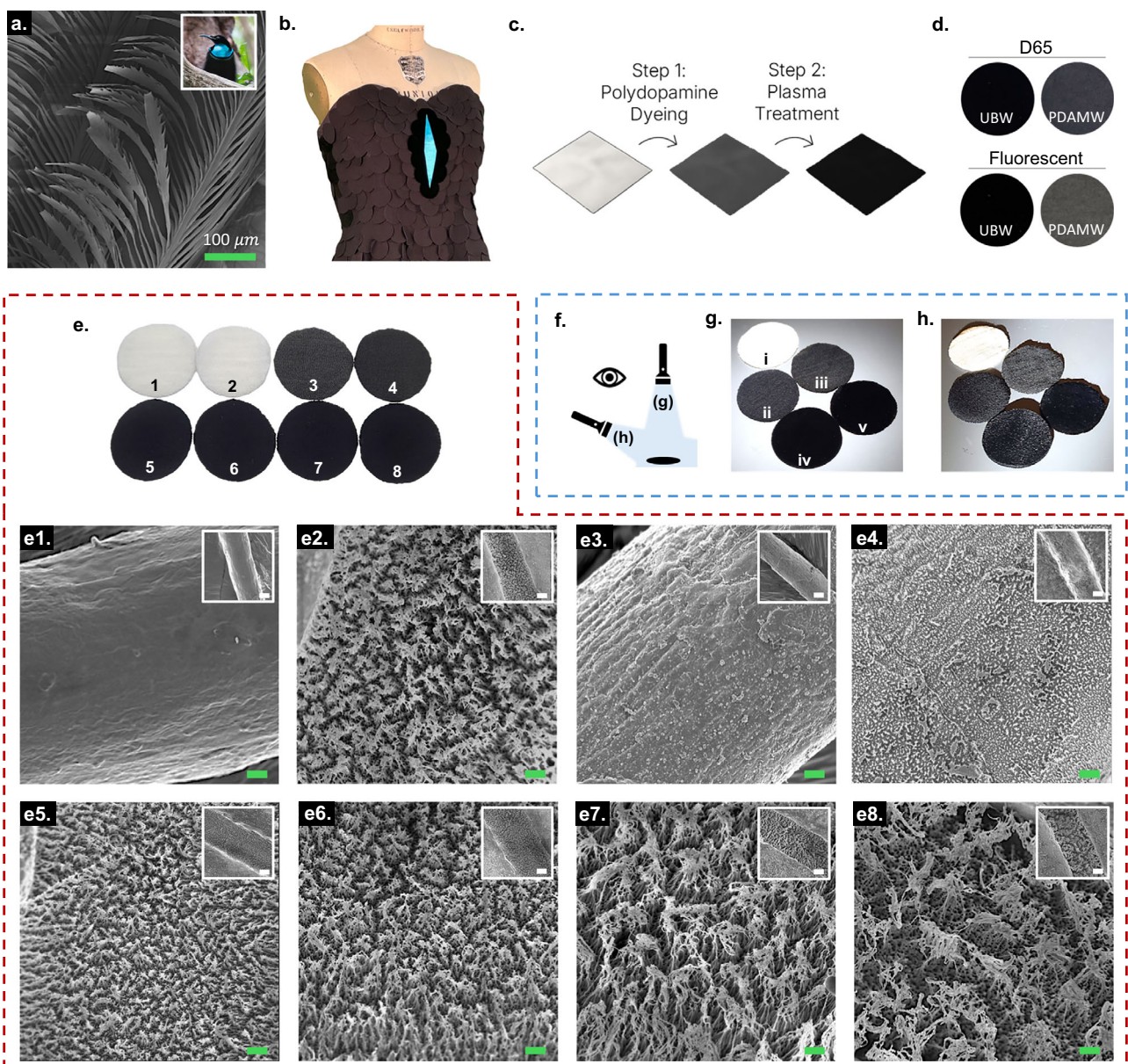

**Fig. 1 | Visual and microscopic surface analyses of nano/microstructures.**
**a** Scanning electron microscope (SEM) image of barbs/barbules from a feather of the magnificent riflebird (*Ptiloris Magnificus*), a species of bird-of-paradise family. Inset image of bird, photo credit: Eric Gropp, CC BY 2.0, https://commons.wikimedia.org/w/index.php?curid=63294234. **b** Black dress inspired by bird-of-paradise, fashioned with black polyester blend (BPETB) and a centerpiece consisting of the ultrablack wool (UBW). **c** Schematic of two-step method to create UBW. **d** UBW and the control fabric, PDA-dyed merino wool (PDAMW), under two different standard lighting systems. **e** Photographic image of (1) white merino wool (WMW), (2) plasma treated WMW (40 W 80 min), (3) PDAMW, and (4–8) plasma treated PDAMW (40 W 5, 30, 50, 80, and 110 min, respectively) (Diameter of samples: 38 mm) under standard lighting system with corresponding SEM images (e1-e8). Scale bars = 1 μm; Inset scale bars = 5 μm. **f** Schematic for two directions of tactical flashlight in dark room. **g** Flashlight illuminating from directly ~30 cm above the fabrics (i: WMW, ii: BMW, iii: PDAMW, iv: flock, and v: UBW). **h** Flashlight illuminating from ~60° slanted angle to ~30 cm away from the fabrics. (Diameter of samples: 38 mm).

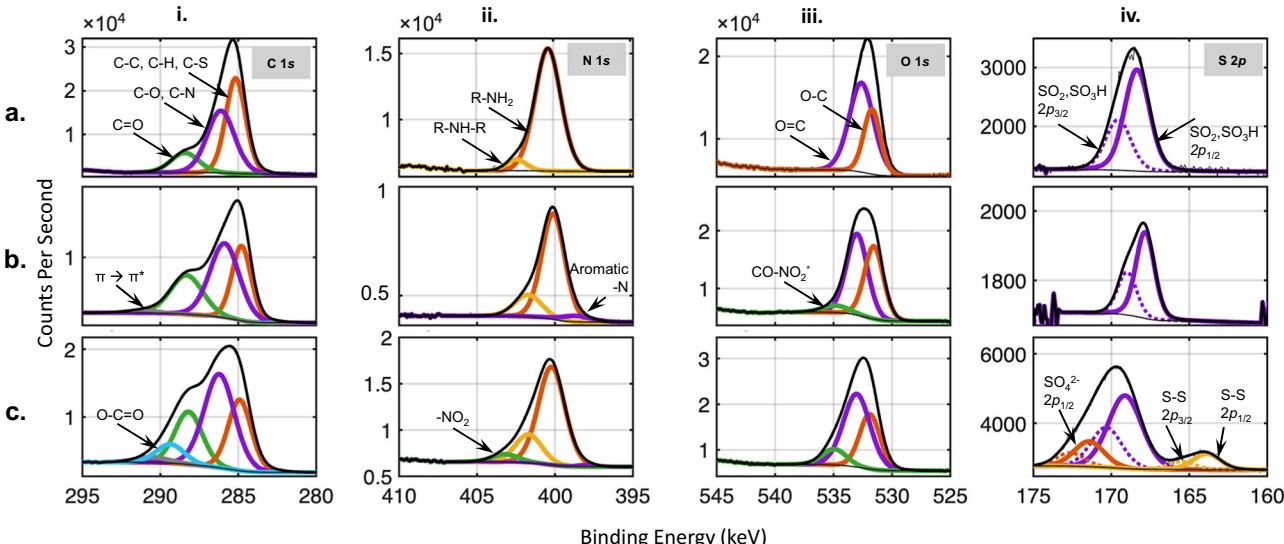

**Fig. 2 | High resolution x-ray photoelectron spectroscopy (XPS) spectra. a** WMW, **b** PDAMW and **c** UBW. For each sample, spectra are recorded for **i.** C 1*s*, **ii.** N 1*s*, **iii.** O 1*s* and **iv.** S 2*p*. The S 2*p* spectra are resolved, including the spin-orbital splitting, 2$p_{3/2}$ (dash lines) and 2$p_{1/2}$ (solid lines).

coloration to WMW, we used PDA. This choice was based on PDA's structural similarity to the melanin pigments that are responsible for ultrablack appearance in nature[1,4,5,17].

To showcase our fabric's ultrablack properties, we provide a side-by-side comparison with other fabrics (see Experimental Section; Fig. 1d, e, Supplementary Fig. 1, and Supplementary Movie 1). We also show the corresponding microstructures of these materials via scanning electron microscopy (SEM; Fig. 1e) to highlight how they contribute to their coloration. In addition to its extreme darkness, our UBW is the only material that remains ultrablack at wide angles of incidence, demonstrating excellent angle-independence when compared to other materials such as commercially available ultrablack materials, black merino wool (BMW), and PDAMW (Fig. 1g, h).

Similar to the ultrablack bird, the SEM images of plasma etched fabrics (Fig. 1e) show hierarchical structures; however, unlike the microscale external features of the feathers, the etched fabric surfaces show both micro and nano features (Supplementary Fig. 2). Plasma etching creates nano/microstructures on white wool (Fig. 1e2) but produces no visual changes to coloration, even with a plasma etching condition identical to Fig. 1e7, which creates ultrablack conditions in PDAMW. Further, while our plasma treatment conditions produced gradations in structural features (Supplementary Table 1), we found no condition after 30 min plasma treatment time where the etched dyed wool did not express ultrablack reflectance (<0.5%; Supplementary Fig. 3). Consequently, the four fabric samples in the bottom row of Fig. 1e appear nearly identical in darkness under standard lighting.

For simplicity, we refer to the nano- and micro-structures as bundles that are composed of multiple nanofibrils (Supplementary Fig. 4a). We predict that this architecture results from increased etching depth, which creates taller nanofibrils with height (*H*), that promotes stronger secondary interactions, such as van der Waals forces[18], causing nanofibrils to collapse inward to form larger bundles. As etching progresses, the complete removal of nanofibrils also results in the formation of nanopores on the surface of the fiber (with diameters, *D*). As etching time increases, structural anisotropy occurs, which is characterized by increasing width of bundles (*W*), *H*, as well as the periodic gap between bundles or the reduced bundle density per unit surface area (Fig. 1e3-e8 and Supplementary Fig. 4b).

## Table 1 | Atomic concentrations (%) of WMW, etched WMW, PDAMW, and UBW

| | Atomic concentration (%) | | | | | | |
|---|---|---|---|---|---|---|---|
| | **C** | **N** | **O** | **S** | **Al** | **N/C** | **C/O** |
| WMW | 75.56 | 8.29 | 15.20 | 0.96 | - | 0.110 | 4.98 |
| Etched WMW | 45.20 | 10.07 | 35.62 | 4.81 | 4.29 | 0.223 | 1.27 |
| PDAMW | 65.48 | 8.45 | 25.89 | 0.18 | - | 0.129 | 2.53 |
| UBW | 51.33 | 12.34 | 28.67 | 2.17 | 5.49 | 0.240 | 1.79 |

## Chemical analysis of PDA dyed and plasma etched wool

To create our ultrablack textile, it is necessary to dye the fabric by embedding PDA within the fiber, rather than simply coating the surfaces. In this study, aggregation and subsequent deposition of PDA particles are controlled by using sodium periodate salt that enables rapid polymerization while improving the hydrophilicity of the solution (pH < 5)[19,20]. Complete dye penetration is confirmed from fabric cross-section images (Supplementary Fig. 5a) and the initial PDA nanoparticles observed on the dyed surface (Fig 1e3 and Supplementary Fig. 5b) are etched out at plasma exposure time <5 min (Fig. 1e4 and Supplementary Fig. 5c).

We use x-ray photoelectron spectroscopy (XPS) and fourier transform infrared spectroscopy (FTIR) to analyze the surface chemistry and to confirm the role of wool and PDA in creating the UBW fabric (Fig. 2, Supplementary Table 2 and 3 and Supplementary Fig. 6). The increased N/C ratio and S content (Table 1) of etched fabrics compared to unetched fabrics suggest the exposure of the proteinaceous structure by removal of the lipid monolayer of the wool epicuticle membrane[21]. Furthermore, according to previous studies, the lower carbon content of PDAMW compared to WMW is more favorable for plasma etching[22]. Our SEM images (Fig 1e2-e7) confirm that the UBW experiences greater etching depths than WMW for the same exposure times, albeit with similar structure. Therefore, PDA facilitates effective surface etching, in addition to having complex polymeric structures with multiple absorption features over a broad range of wavelengths[23-25]. Furthermore, we deconvolute the high-resolution photoemission peaks of C 1s, N 1s, O 1s, and S 2p to confirm the plasma-induced surface oxidation of UBW and the presence of PDA in PDAMW and UBW (Fig. 2). More detailed analysis is provided in Supplementary Table 3.

XPS data also reveal aluminum on etched WMW and UBW (Table 1). To further understand the role of aluminum for anisotropic etching, we perform energy-dispersive x-ray spectroscopy (EDX) analysis. While methods like lithography and metallic mask co-deposition are used for anisotropic etching[22,26–29], aluminum is only present on the non-etched bundle tops, and not in the etched valleys (Supplementary Fig. 7). Since the fabric was not masked, this suggests aluminum, released from the plasma chamber's interior walls and cathode plate, being redeposited onto the fiber surface, disrupting uniform etching. As a result, non-homogeneous etching occurs, leaving behind nanofibrillar structures.

## Spectrophotometric analysis

The lightness of a fabric is quantified by its $L^*$ value on the CIELAB spectrophotometer scale (0–100), where 0 is pure black and 100 is pure white. Plasma etching time directly influences the $L^*$ value of fabrics. We achieve ultrablack fabrics ($L^* <5$, corresponding to %R < 0.5; Supplementary Fig. 3a) with etching times ranging from 30 to 110 min (Fig. 1e5-e8). $L^*$ values for samples in Fig. 1e were measured as follows: 93.72 ± 0.24 (WMW), 93.71 ± 0.18 (WMW, 40 W, 80 min), 13.60 ± 0.40 (PDAMW), 9.59 ± 0.34 (40 W, 5 min), 1.53 ± 0.28 (40 W, 30 min), 1.03 ± 0.20 (40 W, 50 min), 0.69 ± 0.16 (40 W, 80 min), and 1.09 ± 0.42 (40 W, 110 min), respectively. To demonstrate the ease and reliability of achieving an ultrablack state, we also tested other plasma conditions (i.e., plasma power ranging from 20 to 150 W, Supplementary Fig. 3b). Compared to flock fabric, the darkest commercially available fabric-like material, that we observe to have $L^*$ value as low as 1.06, UBW achieves a $L^*$ of 0.50 under the 40 W, 80 min condition. Such extreme darkness in fabric has neither been reported in prior studies nor for purchase commercially.

To accurately represent and reproduce the measured colors, we converted measured-CIELAB values to HEX and RGB codes (Fig. 3a and Supplementary Table 4). UBW shows the lowest color intensity across all schemes when compared to commercial composite fabrics, previously reported fabrics, and control PDAMW. In Fig. 3a, UBW (labeled "A") has RGB values of (2, 2, 2), indicating near-zero intensity in the scale of 0 to 255 and an appearance that is virtually black.

In Fig. 3b, the total reflectance (%R), total transmittance (%T), and total absorbance (%A) of four fabric samples in the visible spectrum ($\lambda = 400$–700 nm) are shown (Full spectrum in Supplementary Fig. 8). Compared to control, PDAMW, UBW's nanostructured surface significantly enhances light-trapping. Despite the two fabrics differing only in surface nanostructures, UBW shows significantly reduced %R and %T, leading to increased %A. Both UBW and flock fabric (Musou Kiwami) qualify as ultrablack with reflectance below 0.5% across the visible spectrum. Ultrablack flocked fabrics, commonly used as photographic backdrops, consist of forest-like vertically oriented monofilament rayon fibers bonded to a polyurethane composite base. This composite structure further suppresses light transmission but compromises flexibility, breathability, and drape—key features of conventional fabrics. SEM images reveal clear structural contrasts between knit wool and flocked fabrics (Supplementary Fig. 9), supporting the explanation for why wool fabrics show higher %T than the flock. Wool and flock fabrics measured approximately 0.7 mm and 1.7 mm thick, respectively, with corresponding differences in grams per square meter (GSM) and volume (Fig. 3c). Beyond the inherent advantages of knit fabric construction, UBW surpasses flock fabric in total reflectance (%R$_{avg}$ = 0.130 vs. 0.274; Fig. 3c). Such darkness exceeds that of the bird-of-paradise plumage (%R$_{avg}$ = 0.97), though the specimen's barbs may not have been optimally oriented as they would be in a live bird (Supplementary Fig. 10)[2].

Transflectance (%TR), defined as the sum of reflectance and transmittance, serves two key purposes. First, it highlights the reduction in both %R and %T for UBW compared to other wool fabrics without additional materials (Fig. 3d). Second, it provides a more comprehensive assessment of the ultrablack angular stability, demonstrating its angle-independence and avoiding the reflectance bias noted in the bird-of-paradise study[2] (Fig. 3e). As shown in Fig. 3d, all three wool fabrics exhibit symmetric responses, but UBW shows a sharp decrease in %TR with tilt, unlike the other two, which display only a mild reduction. At higher tilt angles, UBW's %TR drops more significantly. In contrast, Fig. 3e shows that the flock fabric displays an asymmetric response. This inconsistent optical behavior corresponds to an angle-dependent reflectance bias, in this case, with lowest %TR around +20° tilt and increased %TR at −20°, +50°, and +60°. This translates to the flock fabric appearing darkest at slight tilts, but becoming much brighter at other off-center angles, as shown in Fig. 1h and additional images in Supplementary Fig. 11. UBW, however, is considered angle-independent, as it shows a low transflectance at all probed angles. The higher %TR near 0° is primarily due to relatively high %T (Fig. 3b, Supplementary Fig. 9), which naturally decreases with increased tilt, as long as %R remains low—consistent with the trend observed in Figs. 3b, e. An additional experiment was done to isolate the UBW's surface reflectance from the %T caused by the fabric's open structure. By placing the UBW on a zero-transmittance backing, we directly measure its %R and %TR, and the results confirm the material's wide-angle ultrablack performance. (Supplementary Fig. 12) Lastly, in addition, the near-infrared (NIR) absorbance property of PDA (Supplementary Fig. 8) causes UBW to appear significantly darker than other fabrics under a night-vision camera, while no significance is observed from thermal imaging (Supplementary Figs. 13–17).

## FDTD modeling of nano/microstructures

From our SEM images (Fig. 1e5–e8), we observe: (i) the formation of bundles from collapsed nanofibrils, (ii) changes in the bundle's height ($H$) and width ($W$) with etching time, and (iii) reduced nanofibril density (by complete removal of nanofibrils) and formation of nanopores. Therefore, to obtain a comprehensive understanding of the structure induced optical phenomena of UBW, we simulate the spatial electric field ($|E|$) distribution of nanofibrils ($F$) and bundles ($B$) on a single fiber and collect average reflectance, %R$_{avg}$ ($\lambda = 400$–700 nm; Fig. 4a–d and Table 2) and investigate the optical effects from our observations. The dimensions used in these simulations are identified in Fig. 4a and Supplementary Fig. 4.

We see no significant effect on the %R$_{avg}$ of fibrils ($F_1$ to $F_2$) or bundles ($B_1$ to $B_2$) when increasing the aspect ratio from 1:20 to 1:50; however, the fibrils consistently showed a lower %R$_{avg}$ than the bundles (Table 2 and Fig. 4ci–ii). The $|E|$ distribution suggests that the nanofibrils form a thick forest that periodically traps light between fibrils (Fig. 4bi–iv and 4d) whereas the bundles display nonuniform $|E|$ distribution with optimum light trapping occurring at the core (Fig. 4bv–vi and 4d). With increasing the gap between nanofibrils, $G$, there is a decrease in %R$_{avg}$ of fibrils ($F_2$ to $F_4$), but an increase in %R$_{avg}$ for the bundles ($B_2$ to $B_4$; Table 2). These results suggest a correlation between feature type and its dimensions to the optical property[16,30,31].

In our simulations, the most significant drop in %R$_{avg}$ occurs with the incorporation of nanopores with bundles ($B_3$ to $B_3P_O$, $B_4$ to $B_4P_O$, $B_5$ to $B_5P_O$ and $B_6$ to $B_6P_O$; Table 2, Fig. 4civ and 4d). We surmise the nanopores reduce the %R$_{avg}$ by suppressing the light from escaping from the bundles by increasing the internal scattering. In contrast to bundles with nanopores, nanofibrils incorporated with nanopores increases the reflectivity ($F_3$ to $F_3P_O$ and $F_4$ to $F_4P_O$) as the open-end structure allows light to escape (Fig. 4biii–iv, 4c iii and 4d, e). We therefore conclude the microscale bundles with nanopores are necessary to effectively create ultrablack fabrics from collapsed nanostructures, while the same cannot be said for low aspect ratio (1:20 or 1:50) open-ended nanofibril forests with a nanopore base (Fig. 4e).

Finally, to understand the experimental $L^*$ trend observed as etching time increases, we compare the models $B_O$, $B_3P_O$, and $B_5P_O$

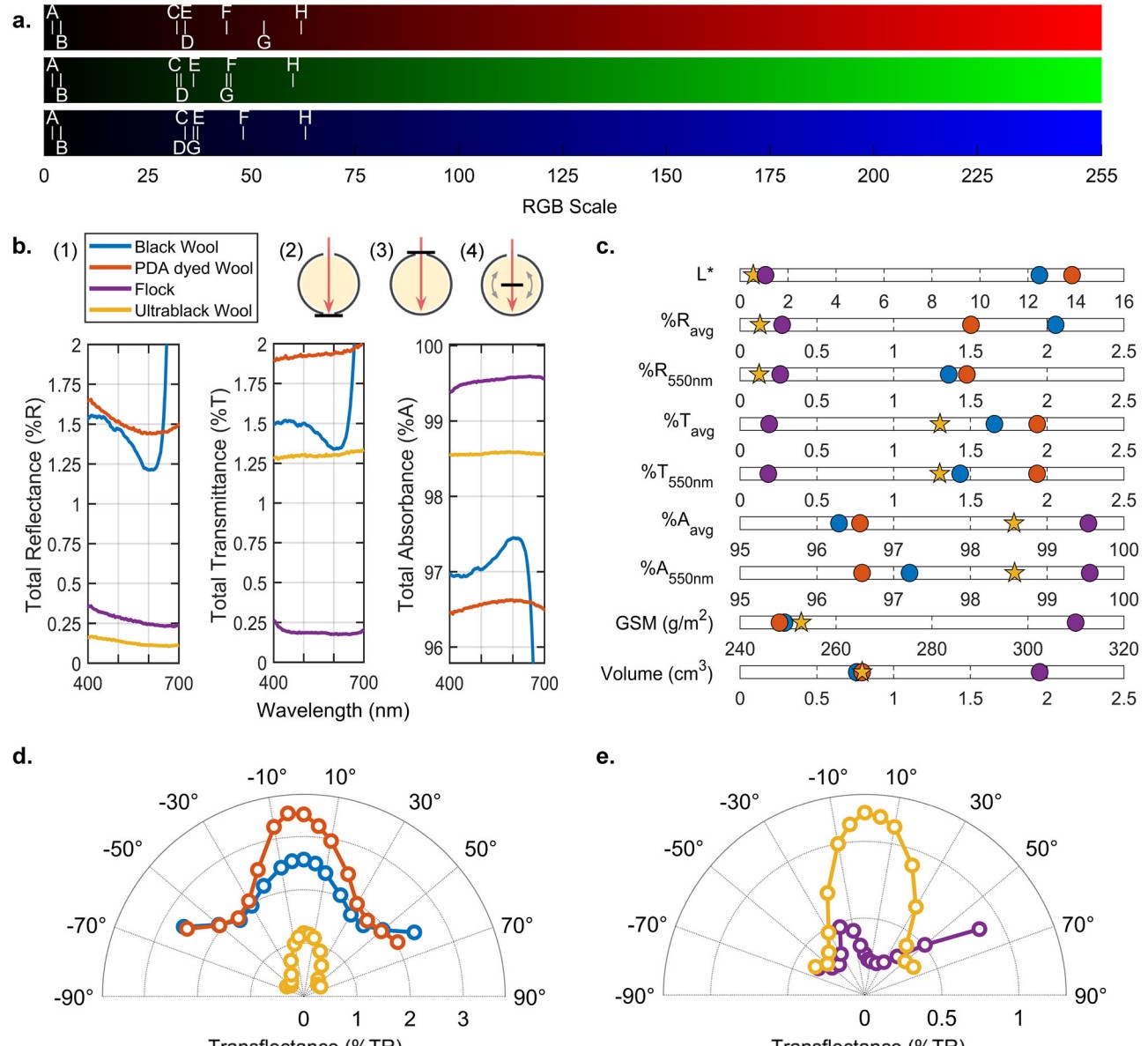

**Fig. 3 | Spectrophotometric analysis. a** Red, green, and blue (RGB) scaling system representing the intensity of each color in corresponding samples (Fabric samples: A. UBW, B. Flock, C. Black merino wool (BMW), D. Black polyester blend (BPETB), E. PDAMW, F. Dark gray polyester (DGPET), G. PDA-dyed Eri silk (PDAES)[32], H. PDA-dyed silk (PDAS)[33]). **b** Total reflectance (%R), total transmittance (%T), and total absorbance (%A) within the visible spectrum ($\lambda = 400$–$700$ nm) with (1) legend for Fig. 3b–e, and schematic illustrations for (2) total reflectance, (3) total transmittance, and (4) transflectance (%TR). **c** Graphical summary comparing L* value, GSM (g/m$^2$), volume (cm$^3$), and average and wavelength specific ($\lambda = 550$ nm) %R, %T, and %A. **d** Angle variable transflectance of wool samples. **e** Angle variable transflectance of UBW and commercially available ultrablack flocked fabric.

(Table 2). Based on SEM images, $H$ and $W$ of bundles increase with etching time (Supplementary Fig. 4 and Supplementary Table 1). As we have noted from our experimental results, initially, bundles with relatively low aspect ratios are formed ($B_0$). As etching progresses, $H$ increases and nanopores form, leading to a drop in %$R_{avg}$ ($B_0$ to $B_3P_0$). With continued etching ($B_3P_0$ to $B_5P_0$), however, %$R_{avg}$ further reduces as $W$ increases with the formation of larger bundles. While the aspect ratio plays a role, we attribute the non-linear, but progressively decreasing L* trend observed with etching up to 80 min to the absolute values of $H$ and $W$, which have a critical effect on %$R_{avg}$. Simplified simulated structures, regular periodicity, and the optical constants ($n$ and $K$) of keratin[3] used to define the material (Supplementary Fig. 18), result in the numerical %$R_{avg}$ discrepancy, but the simulation results align well with the spectroscopic experimental trends (Fig. 3). In creating the UBW fabric, we therefore emphasize the importance of

surface morphologies, their dimensions, as well as their assembly for their optical behavior.

## Textile properties of the UBW

Assessing the textile properties of fabrics is important to confirm their usage in day-to-day applications. Table 3 summarizes the wash fastness, lightfastness, and mechanical strength of UBW compared to PDAMW. The PDAMW has wash fastness similar to PDAES, PDAS, and PDAW[32–34], while the L* of UBW increases by +4.13 on accelerated washing. With exposure to light, the L* value of UBW increases by +0.77, an indistinct color change when compared to PDAMW, which shows a more visible color change. PDA dyeing also increases the tensile strength of the yarn by 0.01 MPa due to the chemical interaction between PDA and wool[35,36], while the subsequent surface level etching does not affect the tensile strength. We also report the

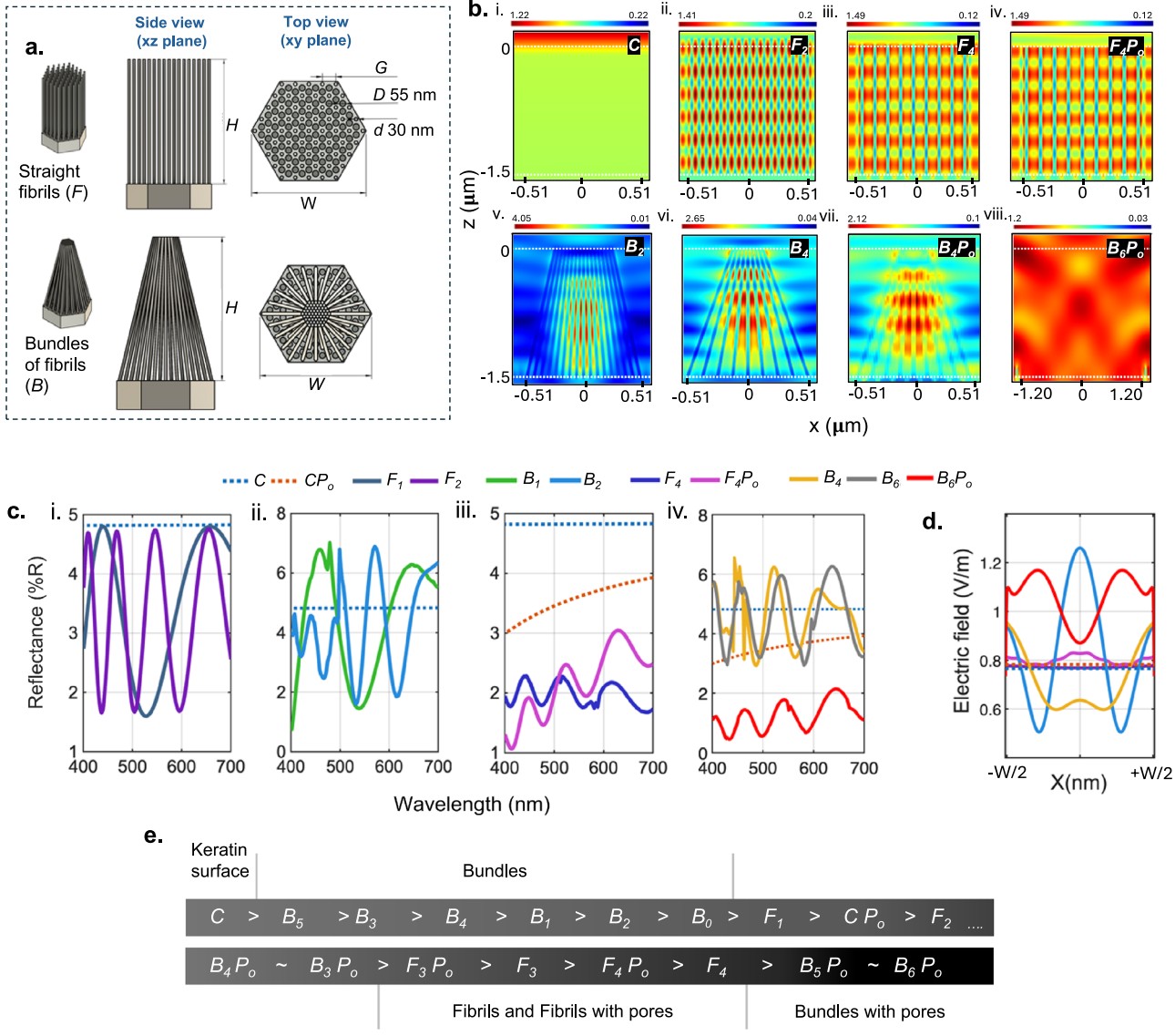

**Fig. 4 | Finite difference time domain (FDTD) modeling. a** Schematic of the side view and top view of the nanofibrils (*F*) and bundle (*B*) are illustrated, and dimensions are marked. **b** Spatial electric field distributions of TM polarized light of wavelength 550 nm. **c** Total reflectance for models i. $F_1$ vs $F_2$ (increasing aspect ratio of *F*) ii. $B_1$ vs $B_2$ (increasing aspect ratio of *B*), iii. $F_4$ vs $F_4P_o$ (incorporating nanopores (*D* -55 nm) with *F*) and iv. $B_6$ vs $B_6P_o$ (incorporating nanopores (*D* -55 nm) with *B*). **d** Electric field distribution of the models depicting the effect of incorporating nanopores to *F* and *B*. The dotted lines represent a smooth keratin surface (blue) and keratin surface with nanopores (orange). **e** Structural comparison of the models by arranging them in decreasing order of %R$_{avg}$.

spectroscopic analysis of a compressed UBW fabric (compressed at 2370 kPa for 4 min), confirming minor impact (Supplementary Fig. 19 and Supplementary Table 5). To provide a direct comparison, identical mechanical load was applied to flock fabric and showed a distinctly different outcome (Supplementary Fig. 20), highlighting the UBW's robust optical symmetry under stress. In summary, UBW remains uncompromised as a woolen textile, and with its production, presents itself as an economical option to produce an ultrablack textile (Supplementary Tables 6 and 7).

In summary, we have introduced a simple and environmentally friendly two-step method to create wide-angle ultrablack wearable textiles. This phenomenon is the result of anisotropic etching of the wool fabrics, which creates bundles of nanofibrils on yarn surfaces. These nanofibrils decrease the Fresnel reflectance, which ultimately results in effective light trapping. We assessed the structural and dimensional effects of the nanostructure by microscopic analyses and simulations. This work highlights the versatility of our method

by providing multiple conditions to achieve ultrablack fabrics. Starting with a natural, conventional knit fabric and fabricating the bundles of nanofibrils with a dry plasma etching method, the UBW from this study achieves the desirable properties of an everyday textile while also achieving the status of the darkest known wearable fabric, even when compared to commercially available flock fabric, which was previously known to have the lowest light reflectance. UBW not only outperforms commercial flock in the total reflectance, but also in retaining its symmetric darkness across a 120° wide angular span.

Our research signifies a paradigm of soft material being used for optical applications spanning from aesthetic implementations and decorations to camouflages and specialized light sensitive apparatuses. We expect that such a simple approach in the fabrication of nanostructures on textiles and soft materials will have pivotal implications in interdisciplinary fields, including textile engineering, polymer science, photonics, and nanotechnology.

**Table 2 | Dimensions of the models used for the optical simulations and the recorded %R$_{avg}$**

| Sample | C | C P$_o$ | F$_1$ | F$_2$ | F$_3$ | F$_4$ | F$_4$ P$_o$ | F$_3$ P$_o$ | B$_0$ | B$_1$ | B$_2$ | B$_3$ | B$_4$ | B$_5$ | B$_6$ | B$_3$ P$_o$ | B$_4$ P$_o$ | B$_5$ P$_o$ | B$_6$ P$_o$ |
|---|---|---|---|---|---|---|---|---|---|---|---|---|---|---|---|---|---|---|---|
| Height (μm), H | - | - | 0.6 | 1.5 | 0.6 | 1.5 | 1.5 | 0.6 | 0.3 | 0.6 | 1.5 | 0.6 | 1.5 | 0.6 | 1.5 | 0.6 | 1.5 | 0.6 | 1.5 |
| Bundle width (μm), W | - | - | - | - | - | - | - | - | 0.78 | 1.02 | 1.02 | 1.02 | 1.02 | 2.04 | 2.04 | 1.02 | 1.02 | 2.04 | 2.04 |
| Gap between fibrils (nm), G | - | - | 60 | 60 | 120 | 120 | 120 | 120 | 60 | 60 | 60 | 120 | 120 | 120 | 120 | 120 | 120 | 120 | 120 |
| %R$_{avg}$ | 4.82 | 3.48 | 3.5 | 3.33 | 2.06 | 1.98 | 2.02 | 2.12 | 3.78 | 4.35 | 4.1 | 4.72 | 4.63 | 4.75 | 4.56 | 2.51 | 2.58 | 1.17 | 1.17 |

Structures include a keratin surface (C), straight fibrils (F) of diameter (d) 30 nm, Bundles of fibrils (B) and, pores (P$_o$) of diameter (D) 55 nm.

**Table 3 | Comparison of textile properties of undyed wool, PDA dyed wool (PDAMW) and ultrablack wool (UBW)**

| Sample | Light fastness (ΔL*) | Wash fastness (ΔL*) | Tensile strength (MPa) |
|---|---|---|---|
| PDAMW | +3.34 | +0.32 | 0.13 ± 0.03 |
| UBW | +0.77 | +4.13 | 0.13 ± 0.01 |

## Methods

### Materials
Undyed and black single jersey merino wool fabrics (200 g/m²) were purchased from Nature's Fabric, USA. Specimen of a Magnificent Riflebird (*Ptiloris Magnificus*) plumage was loaned from Cornell University Museum of Vertebrates (CUMV). Black flocked fabric (Musou Black Fabric Kiwami, Cupro base/Rayon pile) was purchased from Musou Black, USA. Dopamine hydrochloride (≥98%) and sodium periodate (≥99.8%) were purchased from Sigma-Aldrich. All materials and chemicals were used as received.

### One step merino wool dyeing
A solution of sodium periodate (SP, 1 g/L) and dopamine hydrochloride (2 g/L) (molar ratio 1: 2.25) was prepared. Undyed merino wool fabrics (material to liquor ratio of 1:200) were then immersed in the solution and allowed to react at 35 °C for 2 h. The temperature, MRL, and dyeing time were modified from the method given in literature[34], to achieve the optimum level of dyeing. Then the dyed fabrics were washed in water and oven-dried at 40 °C. A fabric sample cutter (38 mm diameter, SDL Atlas) was used to prepare identical diameters of fabric samples for the plasma treatment and characterizations.

### Fabrication of nanostructured surface
PE-100 benchtop RF plasma system (Plasma Etch, Inc.) was used to create nanofibril structures on the surface of PDA dyed wool fabrics. Air plasma was initiated at a chamber pressure of 0.1 Torr without a continuous gas flow. We varied the treatment time and power from 5 min to 2 h and 30 W to 150 W, respectively.

### Finite difference time domain (FDTD) modeling of the nanostructured surface
The surface nanostructure was theoretically simulated by FDTD algorithm using Ansys Lumerical FDTD solutions (2024 R2.1 and 2025 R2). The 3D fiber surface models were constructed using Autodesk Fusion (2025). The reflectance spectrum and electric field distribution (|E|) for wavelengths 400–700 nm were obtained assuming the nanostructures to be periodic, regular, and defect free. In the horizontal direction the simulation field was extended to the infinity with a periodic boundary condition (BC), and the BC in the vertical direction was set as absorbing (perfectly matched layer, PML). A plane wave source with TM polarization was used with reflectance, transmittance, and |E| profile monitors. Simulation parameters used are summarized below. These parameters were kept constant. Also refer reference for setup files and raw data[37].

| Incident medium | Air |
|---|---|
| Material matrix | Keratin |
| Simulation time (fs) | 1000 |
| Pulse duration (fs) | 2 |
| Mesh size (nm) | 5 |
| Mesh accuracy | 2 |
| Distance from reflectance plane to source (nm) | 80 |
| Distance from reflectance plane to structure (nm) | 250 |

## CIELAB spectrophotometer

Colorimetric data were measured using a Ci7800 Sphere Benchtop Spectrophotometer. The instrument simultaneously measures spectral reflectance from 360 to 750 nm at 10 nm intervals. The aperture diameter of the measuring port was 17 mm. Color coordinates were measured according to the CIE L*$a$*$b$*. Illuminating and viewing conditions were chosen to be CIE diffuse/8° geometry.

## Visual analysis

For visual comparison, D65 (daylight) and TL84 (store light, cool white fluorescent) lighting from GTI MiniMatcher were used to evaluate the samples under a standardized lighting system (following the general guidelines of AATCC EP9 "Evaluation Procedure for Visual Assessment of Color Difference of Textiles"). As an additional evaluation method, RBG and Hex digital codes for exemplary samples were generated from L*, a*, to b* values measured using CIELAB spectrophotometer or using the known values. Angle-dependent light absorbance of samples was visualized using a tactical flashlight. CIELAB reporting, HEX code, and RGB scale conversion, and UV-Vis-NIR spectroscopy measurements were based on the UBW sample that had L* of 0.55.

## UV-Vis-NIR spectrophotometer

Total reflectance (%R) and total transmittance (%T) were measured using Agilent Cary 5000 UV-Vis-NIR spectrophotometer with a 150 mm sphere integrating sphere (Cary External DRA) to capture total hemispherical radiation (8° tilt). Total absorbance (%A) was calculated with a simple formula of %A = 100 − (%R + %T) as a conservation of energy. The transflectance (%TR) was measured by mounting a clip-style variable-angle center-mount accessory to the DRA, replacing the original cap from the top. The sample was placed in the center of the integrating sphere facing the incident angle at 0° position (i.e., normal to the sample). As indicated on the top of the center mount module, turning counterclockwise represented a negative angle and turning clockwise represented a positive angle, resulting in the full range of −60° to 0° to 60°.

## Fourier-transform infrared (FT-IR) spectrometer

The functional group analysis of the fabrics was conducted using an attenuated total reflectance Fourier-transform infrared (ATR FTIR) spectrometer (PerkinElmer). The spectra were recorded in the range of 4000–800 cm$^{-1}$ at a resolution of 4 cm$^{-1}$, and 32 scans were conducted for each spectrum.

## X-ray photoelectron spectroscopy (XPS)

X-ray photoelectron spectroscopy (XPS) spectra were collected using Nexsa G2 Surface Analysis System (ThermoFisher Scientific) using a focused Al K monochromatic X-ray source (1486.6 eV). All XPS spectra were analyzed using the CasaXPS software (version 2.3.26) with binding energies charge-corrected to the C 1$s$ peak set at 284.8 eV. Atomic Concentration (%) was calculated using high resolution scans.

## Scanning electron microscopy (SEM) and energy dispersive x-ray (EDX)

The field emission scanning electron microscopy (FE-SEM) (Zeiss Gemini500 FE-SEM) was used to image the wool fabrics after sputter coating with gold-palladium for 40–60 s. For optimum imaging an acceleration voltage of 0.75–1 kV was used. The Energy dispersive x-ray (EDX) spectroscopy was used in conjunction to confirm the elemental distribution on the fabric surface. The acceleration voltage was increased to 5–7 kV. Spot measurements were taken to ensure that the wool surface was not damaged during long mapping measurements and to ensure accurate data was recorded.

## Fabric property assessment

Wash fastness was measured by the standard AATCC61-1A method (accelerated washing method equaling to 5 hand washes). The washing was conducted for 45 min at 40 °C using a laundromat (SDL Atlas Rotawash), rinsed with cold water, air dried, and analyzed using CIELAB spectrophotometer. Light fastness test AATCC 16 - 2004 was performed with the xenon arc lamp equipped light fastness tester (TesttexTF420). Blue reference fabric was used as the control. The fabrics were exposed to light for 42 h. The tensile strength and extension properties of the yarns unraveled from the treated fabrics were assessed using the Instron tester (Instron 5566). The standard ASTM 52256 was modified, 20 cm gauge was used, and extension rate was adjusted such that the breaking occurred at ~20 s. The effect of compression on the nanostructure was assessed using the compressive loading setup of the Instron tester (Instron 5566). Two steel plates (15 ×15 cm) were fitted to the static load cell of 10 kN. The fabric is then placed on a steel disk (diameter 38 mm and thickness 1/8") secured to the bottom plate. The load was gradually increased by vertically moving down the top plate at a rate of 0.5 mm/min until the set load was reached. The fabric was exposed to this load for 4 min before removing the load and taking the CIELAB L* measurement. The above steps were repeated for loads from ~430 kPa to ~2600 kPa. Average of 4 measurements was taken for each load and reported as a plot L* vs. load.

## Data availability

Simulation setups and relevant raw data generated in this study can be accessed at https://doi.org/10.7298/7mve-5139. All other data needed to evaluate the conclusions in the paper are present in the main text and/or the Supplementary information. Raw data files are available from the corresponding author upon request.

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

## Acknowledgements

We thank the lab managers Mark Pfeifer, Alicia Tripp, Malcolm (Mick) Thomas, and John Grazul of Cornell Center for Materials Research (CCMR) and Xia (Sam) Zeng of Human Centered Design (HCD) for training and assisting with characterization. We thank Vanya Rohwer and Mary Margaret Ferraro of the Cornell University Museum of Vertebrates (CUMV) for the loan of the plumage of Ptiloris Magnificus (Magnificent Riflebird). We thank Mia Bressler for assisting with FDTD modeling and Zoe Alvarez for designing the dress. This work made use of the CCMR and HCD shared instrumentation facility.

## Author contributions

L.M.S. conceived the idea and supervised the study. H.J. and K.P. planned and carried out experiments and analyses. All authors discussed the results and contributed to the final manuscript.

## Competing interests

The authors declare the following competing interests: Patent Applicant: Cornell University. Inventors: Larissa M. Shepherd, Kyuin Park, and Hansadi Jayamaha. 11396-01-US. Filed Provisional Patent. The method and data of plasma treating wool to create an ultrablack textile is covered in the provisional patent and manuscript. The authors declare no other competing interests.
