## [Transparent Peer Review file · Nature Communications]

Ultrablack Wool Textiles Inspired by Hierarchical Avian Structure

Corresponding Author: Dr Larissa Shepherd

Version 0:

Reviewer comments:

Reviewer #1

(Remarks to the Author)

The authors report an interesting strategy that suggests a simple method to produce the darkest ultra-black fabric with a reflectance of only 0.13% in the visible spectrum, inspired by nature. The authors present a two-step method involving PDA dyeing of wool and surface treatment using plasma etching. The ultra-black wool was further characterized by morphological observations, chemical analysis, optical and mechanical tests. Although this study presents the darkest fabric inspired by nature using natural dyes and a dry surface treatment technique, we believe that the overall robustness of the materials and method is somewhat limited for application to fabric surfaces. Therefore, I cannot recommend this study for publication in a journal based on the submitted manuscript. Please refer to the detailed comments below that may help improve the study.

1. Please compare the micro/nano-structures between the one from this work and the avian nanostructures that inspired this work. I guess the ultrablack color is induced by several factors of materials and structures (yarn structure, fiber microstructures, nanostructures). Please identify the individual roles to make the wool surface darkest.

2. The authors may analyze the PDA dye formed on the wool fiber surfaces in details. I think the overall coating duration for 2 hours would not have good adhesion of PDA nanoparticles to the wool fiber surfaces. FTIR analysis for the PDA particles only and the wool fabrics treated by plasma only should be compared in Fig. S1.

3. To enhance the dark color, the authors performed the two step method of PDA dye and subsequent plasma treatment. I think that the plasma treated surface should be explored carefully whether or not it still has the PDA dye after oxygen plasma treatment for longer duration than 30 min. Since PDA, a biopolymer, might be etched away totally for longer duration of plasma etching when it has the less thickness than around sub-micrometers. In fact, in supplementary Fig. 2, the PDA particle density seems very lower, which might not be survived after plasma treatment for such a longer duration for more than 30 min. It will be helpful if the authors provide more photo images and relevant analysis for the original wool, the PDA-coated wool before and after plasma treatment with different treatment durations. Please consider the etch rate on the PDA particles and wools surfaces by the air plasma treatment.

4. As the nanostructures were developed on wool with the plasma treatment time. The aspect ratio of the height over the width of nanopillars would be changed. I think that as increases in the aspect ratio, the black color should be darker. Please discuss the threshold of the structural aspect ratio for the darkest wool.

In addition, what happens when the original wool, without PDA coating, is treated for 110 min under air plasma?

5. Since the plasma etching is anisotropic, it has a certain limitation to form the nanostructures on the non-planar surfaces such as the fiber geometry. The images in the inset of the Figs 1e-h showed the topview. Please showed the side view of the microfibers having nanopillars. As the plasma irradiation is effective only on the top-surface (or the surface facing the plasma irradiation), I guess that the sides or bottom of the wool microfibers have different morphologies or no nanostructures. If so, the untreated side of the microfibers might be exposed when it applied for washing. Thus the term of 'omnidirectional ultrablack' might not be properly used in the work?

6. As the authors discussed, Aluminum was found on the top surfaces of the nanostructures originated from the surroundings. Is there any chance that this metallic component affect the reflectance of the fabric?

7. The authors defined the pillars as the clustered nanofibril structures. Please provide a simple schematic to define the pillars (including nanofibrils) for the height, widths, pitches.

8. Air plasma etching would break the polymeric chains on top of the fiber surface, then the mechanical strength of fibers with nanostructures might be much lower than the original one. Please consider the change in the tensile strength of individual fibers after plasma treatment for 60 min and more treatment times?

Reviewer #2

(Remarks to the Author)

The study introduces a two-step process to develop ultrablack wool textiles inspired by avian nanostructures. While the topic is interesting and has potential applications, there are significant issues in the manuscript that need to be addressed.

The manuscript does not sufficiently highlight its innovation. The comparison with existing studies such as Nano Letters 22.23 (2022): 9343-9350 can be provided.

The use of plasma etching is well-documented. However, the study does not adequately demonstrate how this approach significantly improves upon current ultrablack materials in terms of cost, scalability, or end-use performance.

The study provides limited validation for its proposed applications. Although it mentions potential uses in wearable textiles and optical camouflages, more experimental evidence is required to demonstrate UBW's performance under real-world conditions such as environmental exposure or prolonged use.

Reviewer #3

(Remarks to the Author)

The authors present a simple method for producing wool fabric with low reflectance, including at a wide range of incident angles. I think the simplicity of the approach and the excellent optical performance makes this work significant. In order to facilitate comparison with other work, I think calibration/testing with a low reflectance reference standard should be shown. Also, I think the simulation section is hard to follow and needs to be expanded to make the arguments convincing. Below are several questions that I think should be addressed:

- 1) Can the authors provide photographs with high light intensity and/or longer exposure time to provide some information about the homogeneity?
- 2) Can the authors provide a measurement of a reference standard to provide confidence in the quantitative values of reflectance? I think this is important for repeatability of this work and accurate comparison of quantitative values between different groups.
- 3) Is reflectance or transreflectance more important for applications? If just reflectance, couldn't you just make larger gaps to have more light transmit through?
- 4) For the measurements in Fig. 2f and 2g, is the transreflectance at normal incidence higher simply because of the larger transmission through holes in the fabric? If so, can the authors do this experiment with a thicker patch of wool (or maybe two pieces stacked on top of one another) in order to minimize the transmitted part? I think it is important to isolate the angle-dependent reflectance, because this might be the more relevant thing in an actual application.
- 5) I don't understand how the authors are incorporating the nanofibrils within the pillars. Can the authors provide some more explanation? Are the nanofibrils the same as the pillars? What is meant by "surface coupling effects at the individual nanofibril interfaces"?
- 6) Is what is labeled as D in the SEM in the supplement supposed to be W? And then is D something else? Also, D and d are not labeled in Figure 3.
- 7) From the SEMs, the geometry changes significantly across the sample, so I think the authors need to investigate the impact of geometry parameters on the simulation results.
- 8) Page 9, the magnitude of the force in the compression test doesn't seem relevant by itself. I think the authors should convert this to an approximate average pressure and explain how this approximate value is determined.
- 9) The claim in the abstract that the material shows "an omnidirectional ultrablack property, even after compressive loading..." is not substantiated as far as I can tell. The authors do not show angle-dependent integrating sphere measurements after the compressive load test.

Minor comments

- 10) On page 3, the sentence starting with "A coloration meant..." does not seem grammatically correct.
- 11) The authors should probably add labels to the different patches in 2a and 2b
- 12) Page 11, missing unit on 0.01
- 13) I think Figure 1 should also show the samples without plasma processing to understand the starting structure

Version 1:

Reviewer comments:

Reviewer #1

(Remarks to the Author)

I appreciate Authors' effort on the revision. Most of my concerns have been now addressed. Therefore, I recommend the revised manuscript for publication after minor revisions.

1. As the authors noted in their rebuttal, the avian structure is primarily in the microstructure. Then, it would be better to slightly revise the title. Instead of "avian nanostructure," "hierarchical avian structure" or "avian microstructure" might better describe this study.

2. Can the author clarify whether the PDA dye is present on the surface or the inner surface of the wool fibers after 2 hours of immersion dyeing? Ref. 30 in the revised manuscript suggests that PDA adheres to the surface of the silk yarn when dyed. Since plasma etching is performed on fabrics pre-dyed with PDA for more than 30 minutes, if the PDA layer thickness or penetration depth is thin, the dyed PDA may be completely etched onto the upper surface of the wool fiber, affecting the blackness.

Reviewer #2

(Remarks to the Author)
The revision is acceptable.

Reviewer #3

(Remarks to the Author)

Thank you to the authors for the additional details. There are still a few important points that are critical for articulating the novelty of this work and therefore justifying publication in this journal:

1) In the rebuttal letter, the authors claimed that their goal was to not to trade reflection for transmission (i.e. goal was to maximize absorption). Therefore, the transmittance measurements are most important. However, in terms of TR, the flocked fabric outperforms the UBW at all angles except +50 and +60 degrees (Figure 3e). Can the authors comment on this?

2) Also, if absorption is most important, then wouldn't it make most sense to define "ultrablack" in terms of absorption rather than reflection? If this is done, then the UBW would only be ultrablack from -30 to -60 degree and +30 to +60 degrees, right?

3) To be clear, I understand why the authors originally defined "ultrablack" based on reflectance (namely that this is the definition used in other works), but I think this could be misleading because the absence of transmission could be implied in such a definition. This is why I was wondering if the authors could consider characterizing their material with a backing material. I appreciate that the authors didn't want to "cheat" by putting a backing material, but I think UBW would always be used with some material behind it (please correct me if that is wrong). Would it make sense to characterize with a realistic backing material (perhaps human skin or some other material that might be beneath the UBW)? This would allow the authors to have a realistic metric for the amount of light that would be reflected from UBW in a real-world situation.

4) "it shows a low reflectance at all probed angles"...I believe it should say "transmittance".

5) There seems to be substantial nonuniformity in UBW, which is particularly noticeable when illuminated at off-normal angles (Fig 1h). Can the authors comment on the cause of the nonuniformity? How significant are the changes in lightness across the sample? Also, will this nonuniformity be significant in envisioned applications? Is this non-uniformity more significant after compression?

6) After the compression test, UBW no longer outperforms flocked fabric. If the authors think the mechanical robustness is a significant point, then they should compare compressed flocked fabric to compressed UBW.

7) I do not think the statement of "omnidirectional ultrablack" is substantiated. In the discussion of Figure 3, the authors point out that the material is only confirmed to be ultrablack (defined by authors as $R < 0.5\%$) near normal (from integrating sphere reflectance) and between -60 to -30 degrees and +30 - +60 degrees (from angle-dependent TR). This should be clarified in the abstract so as not to overstate the accomplishment. If the authors want to claim ultrablack performance across the whole range, they will need to decouple T and R somehow. However, given that the authors indicate that TR is the important metric, I am now not sure if isolating R is the right thing to do.

8) From Fig. S18d, after compression, the material is only confirmed to be ultrablack ($R < 0.5\%$) at normal incidence because all TR values presented for the compressed material are above 0.5%. The abstract and text should be revised accordingly.

9) It seems like the flocked fabric has the issue of fibrils not being straight, but the wool seems to have a similar issue (specifically Figure S18d is asymmetric) after mechanical loading. Is it possible that the flocked fabric has seen load already during packaging, shipping, and handling and that is why it isn't good at off-normal angles? From experience with flocked fabric, different regions can exhibit different fibril alignment which can be modified by mechanical load. I think this is an important consideration because only the pristine (i.e. not compressed) UBW shows the superlative properties.

Version 2:

Reviewer comments:

Reviewer #1

(Remarks to the Author)

I appreciate the authors for their effort to revise the manuscript. I think all of my concerns have been addressed now. I would like to recommend this work to be published in the journal.

Reviewer #3

(Remarks to the Author)

The authors have addressed all of my concerns.

Thank you to the authors for the thorough response. I hope the authors agree that the additional experiments helped to clarify and reinforce the contributions of this work.

Author:

We thank the reviewers for the valuable and detailed comments. In response, we have made significant changes including presenting data from more thorough optical simulations, adding a chemical analysis section using XPS, and performing additional experiments to confirm the potential application in camouflage. We also acknowledge that some information in our original manuscript has been misinterpreted. Therefore, we have restructured the manuscript and included new schematics, photographs, and SEM images to improve clarity. We thank the reviewers again for helping us to improve our manuscript through constructive feedback.

Below we provide point by point responses (in **purple**) to each of the comments provided. Also, the changes made in the manuscript and supplementary document are with tracked changes (in red).

Reviewer #1 (Remarks to the Author):

Reviewer #1:

The authors report an interesting strategy that suggests a simple method to produce the darkest ultra-black fabric with a reflectance of only 0.13% in the visible spectrum, inspired by nature. The authors present a two-step method involving PDA dyeing of wool and surface treatment using plasma etching. The ultra-black wool was further characterized by morphological observations, chemical analysis, optical and mechanical tests. Although this study presents the darkest fabric inspired by nature using natural dyes and a dry surface treatment technique, ***we believe that the overall robustness of the materials and method is somewhat limited for application to fabric surfaces.*** Therefore, I cannot recommend this study for publication in a journal based on the submitted manuscript. Please refer to the detailed comments below that may help improve the study.

Author:

We appreciate the reviewer for the positive comments and the valuable feedback provided.

This response is mainly for the above comment in ***bold***. Our study focuses on fabricating and characterizing an ultrablack fabric as a cost-effective and industrially viable alternative for applications requiring flexible, breathable, biocompatible, and biodegradable textile-like materials. The aim is to enable: i) wearability for aesthetic purposes or tactical camouflage, and/or ii) economical photothermal devices, as will be outlined in our planned future publications.

Accordingly, our work does not aim to develop a material or process platform that is adaptable to different material types. Instead, it deliberately focuses on creating a cost-effective ultrablack fabric while preserving its intrinsic textile properties. A cost estimate and comparison with the

state-of-the-art vertically aligned CNTs (VACNTs) and the commercially available darkest fabric are now included in the Table S5 and S6. These confirm that our fabric can be produced at a substantially lower cost than VACNTs, while also outperforming flock fabric optically (R% and angle-independent TR%) and maintaining intrinsic textile properties such as light weight, conformability, and breathability.

Reviewer #1:

Please compare the micro/nano-structures between the one from this work and the avian nanostructures that inspired this work.

Author:

We have added a brief comparison of the length scales of the avian microstructure and fabric structures in the manuscript in Section 2.1. Both structures are hierarchical, while the avian structure is predominantly in the microscale, the textile has both nano and microscale features which enable the ultrablack property and creation of the darkest fabric reported so far. We also compare the two structures in Section 2.3 when mentioning the angle-independent ultrablack behavior. Unlike the bird which has an incident angle bias, the fabric shows an angle independence due to the 3D arrangement of the individual fibers and yarn in the fabric.

Reviewer #1:

I guess the ultrablack color is induced by several factors of materials and structures (yarn structure, fiber microstructures, nanostructures). Please identify the individual roles to make the wool surface darkest.

Author:

We agree that the materials and structures (yarn structure, fiber microstructures, nanostructures) play individual/synergistic roles in creating the angle-independent ultra black effect. Following explanations and experimental data are included in the manuscript:

1. Material effect: The choice of merino wool and polydopamine (PDA) is intended to achieve the ultrablack effect. The reasoning is as follows:

- i. Merino wool fibers behave as a wearable and flexible substrate with chemical functionalities that effectively take up PDA which provides the durable black color (Table S2 and Section 2.2).
- ii. It was realized during preliminary studies that PDA, similar to naturally available melanin structures, can be used to achieve extremely dark substrates. The correlation between the PDA structure and the black color is an area of debate among scientific community.¹ However, given the chemical heterogeneity, PDA possess multiple absorption features over a broad range of wavelengths.¹⁻³ More comprehensive explanation with XPS data is now added to the manuscript Section 2.2 and Table S2; Chemical modification of the wool surface by PDA also facilitate deeper etching compared to a wool fabric without PDA. We also include the SEM image of etched undyed merino wool (Fig. 1e2) to show the clear difference in etching depth. This finding is inline with previous studies, that suggest having a lower carbon content being favorable for etching.³

2. Fiber/yarn/fabric structure effect:

As mentioned in Section 2.3, the three-dimensional arrangement of fibers and yarns in the single jersey knit fabric structure enables an “angle-independent” ultrablack effect, unlike the avian nanostructure, which exhibits anisotropy due to the regular, tilted arrangement of micro-barbules on feathers, resulting in an angle bias. The cross-sectional SEM images now added to the Fig. S2 confirms this – the nanostructure follows the contour of the fiber top surface, naturally creating complex dimensional heights and layers, which allows the nano/microtextured surface to capture light from a range of incident angles.

References:

1. Ball, V. (2017). Determination of the extinction coefficient of “polydopamine” films obtained by using NaIO₄ as the oxidant. *Materials Chemistry and Physics*, 186, 546-551
2. W. Sheng, B. Li, X. Wang, B. Dai, B. Yu, X. Jia and F. Zhou, Brushing up from ‘anywhere’ under sunlight: a universal surface-initiated polymerization from polydopamine-coated surfaces, *Chem. Sci.*, 2015, 6, 2068–2073
3. Qie, R., Moghaddam, S. Z., & Thormann, E. (2021). Parameterization of the optical constants of polydopamine films for spectroscopic ellipsometry studies. *Physical Chemistry Chemical Physics*, 23(9), 5516-5526.

Reviewer #1

2. The authors may analyze the PDA dye formed on the wool fiber surfaces in details. I think the overall coating duration for 2 hours would not have good adhesion of PDA nanoparticles to the wool fiber surfaces.

Author:

The PDA dyed fabrics are formed by dye molecules and/or nanoparticles *penetrating* the fibers, while the low amounts of nanoparticles deposited *on* the fiber surface has minimal impact on the final color, as they are anyway etched out during the plasma process. We have added SEM images of the fabric surface after etching for 5 minutes to confirm this (Fig. 1e4 and Fig. S5).

Even with only 2 hours of dyeing, the strong adhesive property of PDA,⁴ results in good fastness properties. This is confirmed from the wash fastness data available in Table 3, showing no color change after a standard 20 min accelerated wash cycle, consistent with literature.⁵ However, in light of this comment, we revised the text to highlight that the ultrablack effect arises from dye/particle penetration rather than surface nanoparticle or coating formation (Section 2.2).

References:

4. Liu, Y., Ai, K., & Lu, L. (2014). Polydopamine and its derivative materials: synthesis and promising applications in energy, environmental, and biomedical fields. *Chemical reviews*, 114(9), 5057-5115.
 5. Huang, H., Zhang, W., Han, X., Han, Z., Song, D., Li, W., ... & Xu, W. (2023). Effect of polydopamine deposition on wool fibers on the construction of melanin. *Journal of Applied Polymer Science*, 140(5), e53396.
-

Reviewer #1:

FTIR analysis for the PDA particles only and the wool fabrics treated by plasma only should be compared in Fig. S1.

Author:

We have now included the FTIR data for PDA particles in Fig. S6 and included a more extensive analysis.

The appearance of few signature peaks such as the shoulder peak at $\sim 1713\text{ cm}^{-1}$ corresponding to the C=O in quinone confirms the PDA polymerization and presence of PDA in dyed merino wool. Additional characteristic vibrations of PDA include C=C ($\sim 1596\text{ cm}^{-1}$) and C=N ($\sim 1510\text{ cm}^{-1}$) stretching vibrations of the indole ring; however, these signals are overlapped by the broad amide

I band of wool. Therefore, a more extensive chemical analysis is done using XPS which is now included (Figure 2, Section 2.2 in manuscript and Table S2). The analysis not only confirms the presence of PDA but also identifies the importance of PDA and wool in creating an ultrablack fabric.

Reviewer #1:

3. To enhance the dark color, the authors performed the two step method of PDA dye and subsequent plasma treatment. I think that the plasma treated surface should be explored carefully whether or not it still has the PDA dye after oxygen plasma treatment for longer duration than 30 min. Since PDA, a biopolymer, might be etched away totally for longer duration of plasma etching when it has the less thickness than around sub-micrometers. In fact, in supplementary Fig. 2, the PDA particle density seems very lower, which might not be survived after plasma treatment for such a longer duration for more than 30 min. It will be helpful if the authors provide more photo images and relevant analysis for the original wool, the PDA-coated wool before and after plasma treatment with different treatment durations. Please consider the etch rate on the PDA particles and wools surfaces by the air plasma treatment.

Author:

As we noted in our response to Comment 2, our focus is on dyeing the fabric rather than “coating,” although a low density of nanoparticles may be concomitantly deposited during dyeing and subsequently etched away. SEM images below show that PDA nanoparticles are removed after 5 minutes of plasma exposure; at this point, the fiber surface has no visible PDA nanoparticles and shows signs of etching. For clarity, we have included an explanation for the dyeing process (Section 2.2) emphasizing the importance of dye penetration and how it is achieved. As advised by the reviewer, we have also added photographs of the original wool, PDA-dyed wool, and plasma-etched wool (Fig. 1e1-3), and confirmed the presence of PDA in both dyed and plasma-etched fabrics using FTIR (Fig. S6) and XPS (Fig. 2 and Section 2.2).

Figure: **a.** Surface of the PDAMW showing PDA nanoparticles, **b.** Fiber surface after 5 minute plasma etching. The surface has started to etch away and PDA nanoparticles are not observed. Scale bar 1 micron.

Reviewer #1:

4. As the nanostructures were developed on wool with the plasma treatment time. The aspect ratio of the height over the width of nanopillars would be changed. I think that as increases in the aspect ratio, the black color should be darker. Please discuss the threshold of the structural aspect ratio for the darkest wool.

Author:

We thank the reviewer for this thoughtful comment. We agree that the aspect ratio influences the optical behavior of structured surfaces. However, we have realized that the nature of the structural features of the current study demands more intricate studies, and so instead of identifying a threshold aspect ratio, we extended the optical simulations to capture the variables that affect optical behavior. Our thought process is explained below.

Firstly, for clarity we include the above figure (also added to Fig. S4) which differentiate the terms “nanofibrils” and “pillars”. In the manuscript as well as in this response, for simplicity, we have changed the term “pillar” to “bundles of fibrils” or in short “bundles”.

Key observations from the SEM images of the fiber surface at increasing etching time (Fig 1e5 – e8) are:

- i. The height (H) of fibrils/bundles increases with etching time
- ii. The width (W) of bundles increases with the etching time

iii. The width (d) of nanofibrils remain unchanged (~ 30 nm)

First, accurately measuring H is challenging. We tested several approaches, including obtaining fiber cross-sectional SEMs by (i) freeze-fracturing and (ii) embedding in epoxy resin followed by polishing to expose the cross-section (Fig. S2). However, these efforts did not yield images from which dimensions can be accurately measured. These SEM images, however, revealed that H varies significantly along the fiber contour, with the maximum height (H_{\max}) observed at the boundary of the etched surface (Fig. S2g). The H_{\max} can be measured from SEM images taken from fibers directly attached to a SEM stub. While we can calculate the aspect ratio using the average H_{\max} (refer below table), this may not accurately represent the bundle H nor the aspect ratio.

Second, as shown in the table below, as H and W increases with etching time, the aspect ratio changes only slightly since H and W increase simultaneously.

For these two reasons, rather than reporting a threshold aspect ratio for the ultrablack effect, we provide extensive simulations analyzing the effects of individual features (nanofibrils, bundles, and nanopores) and their dimensions, both in isolation and in combination. These simulations confirm that H , W , and their ratio ($W:H$) collectively influence L^* and $R\%$, and that the hierarchical structure is essential for achieving the ultrablack effect.

Plasma time:	30 min	50 min	80 min	110 min
H_{\max} (nm)	1500	1600	3000	3000
W (nm)	450	550	800	1100
$W : H_{\max}$	1:3.3	1:2.9	1:3.8	1:2.7

*Note: the actual H might be smaller, and the resulting aspect ratio also will be smaller than what is reported in the above table.

Reviewer #1:

In addition, what happens when the original wool, without PDA coating, is treated for 110 min under air plasma?

Author:

We appreciate the reviewer's comment, and we have now included the SEM image and photo of plasma treated undyed merino wool in the manuscript (Fig 1e). This confirms that the shape and the hierarchical assembly of nanofibrils into bundles is independent of the presence of PDA and depends on the type of substrate (wool) and the plasma conditions. However, SEM image shows that the etching depth of dyed wool is higher than the undyed wool (Fig 1e). This is explained

using XPS analysis, which confirms the theory that the composition of the substrate has an effect on the nature and extent of etching.⁶ The carbon content being the main factor, PDA dyed wool with lower carbon content than the undyed wool, results in greater etching, this is now mentioned in the manuscript (Section 2.2). Fabric treated for 80 min, which is the optimum etching time for the darkest fabric is used for these analyses.

References:

6. Ko, T. J., Jo, W., Lee, H. J., Oh, K. H., & Moon, M. W. (2015). Nanostructures formed on carbon-based materials with different levels of crystallinity using oxygen plasma treatment. *Thin Solid Films*, 590, 324-329.

Reviewer #1:

5. Since the plasma etching is anisotropic, it has a certain limitation to form the nanostructures on the non-planar surfaces such as the fiber geometry. The images in the inset of the Figs 1e-h showed the top view. Please show the side view of the microfibers having nanopillars. As the plasma irradiation is effective only on the top-surface (or the surface facing the plasma irradiation), I guess that the sides or bottom of the wool microfibers have different morphologies or no nanostructures. If so, the untreated side of the microfibers might be exposed when it applied for washing. Thus, the term of ‘omnidirectional ultrablack’ might not be properly used in the work?

Author:

The reviewer is correct in suggesting that the sides and bottom surface of the wool fibers aren't etched since these surfaces aren't directly exposed to the plasma irradiation (SEM image attached here and added as Fig. S2). During washing, individual fibers can reorient and, in some cases, the unetched fiber surface can face up, however, despite this we see that the L^* value remains ~ 5 suggesting that it is still ultra black. Furthermore, we have now explained the “omnidirectionality” in Section 2.3 and generally, refers to incident angles from $+90^\circ$ to -90° and this doesn't include flipping the fabric surface.

Figure: **a.** UBW fabric embedded in epoxy resin and polished to expose the cross section of the fabric, **b.** Freeze fractured UBW yarn. Both images show the nano/micro textured surface on the top surface of fibers and the surface structure conforming along the fiber top surface arising in a 3-dimensional height and nanofibrillar orientation.

Reviewer #1:

6. As the authors discussed, Aluminum was found on the top surfaces of the nanostructures originated from the surroundings. Is there any chance that this metallic component affect the reflectance of the fabric?

Author:

We appreciate the reviewer's comment regarding the potential effect of aluminum on the reflectance of the fabric. Our EDX/EDS analysis revealed only a relatively low intensity of aluminum at the tips of the nanopillar structures, with no detectable aluminum in the valleys or base regions between the bundles. Had aluminum deposition significantly impacted the fabric's reflectance, this would have been evident in our comprehensive CIELAB, UV-Vis-NIR, and visual analyses.

While investigating the synergistic effects of metallic substances and nanostructure optical properties is a valuable research direction, this falls outside the scope of the current study. We believe that the aluminum deposition from the plasma treatment would be minimal, as the process is fundamentally different from conventional sputtering techniques employing an aluminum target/source. Furthermore, even after gold/palladium sputter coating prior to SEM imaging, the UBW fabric remained unaffected and retained its ultrablack appearance, as illustrated in the photograph below.

This top-down view displays an aluminum SEM stage prepared for imaging. The stage is covered with carbon tape, onto which a small square piece of UBW fabric is mounted. Conductive silver paste was applied to the edges of the UBW to ensure electrical contact with the carbon tape. The prepared stage was then sputter-coated with gold/palladium, which is evident as a silvery appearance on the carbon tape. Notably, the UBW fabric retains its ultrablack appearance despite this sputter coating.

Reviewer #1:

7. The authors defined the pillars as the clustered nanofibril structures. Please provide a simple schematic to define the pillars (including nanofibrils) for the height, widths, pitches.

Author:

We have now included the schematic with the dimensions marked in Fig. S4 and Fig. 4a.

Reviewer #1:

8. Air plasma etching would break the polymeric chains on top of the fiber surface, then the mechanical strength of fibers with nanostructures might be much lower than the original one. Please consider the change in the tensile strength of individual fibers after plasma treatment for 60 min and more treatment times?

Author:

The manuscript includes the tensile strength of plasma treated fibers at optimum etching time of 80 minutes. Although the reviewer's comment is valid, we do not see a change in average tensile strength due to the etching being limited to the outermost (top) surface level of fibers, as evident from the cross-sectional SEM images (Fig. S2). These images also show that the plasma etching is confined solely to the fibers on the outermost surface of the fabric which indicates that the overall mechanical strength of the fabric is not affected.

Reviewer #1 Comments end here

Reviewer #2 (Remarks to the Author):

The study introduces a two-step process to develop ultrablack wool textiles inspired by avian nanostructures. While the topic is interesting and has potential applications, there are significant issues in the manuscript that need to be addressed.

The manuscript does not sufficiently highlight its innovation. The comparison with existing studies such as Nano Letters 22.23 (2022): 9343-9350 can be provided.

Author:

We appreciate the reviewer for the valuable comments provided. We have addressed each comment and adjusted both the manuscript and the supplementary section, accordingly.

We agree that the study utilizing Ppy nanospheres is an excellent example for bio-inspired light-trapping method and appreciate the reviewer for bringing it to our attention. Our study, however, employs a fundamentally different approach.

Our choice of a comparative benchmark, however, was driven by two key factors: structural relevance and performance. Inspired by the feathers of super black bird-of-paradise, which closely resembles the light-trapping mechanism of VACNTs, we chose to benchmark our UBW against the commercially available “flock fabric” for its widely recognized standard as an ultrablack textile with light-trapping structure comprising vertically oriented filaments.

Our UBW fabric achieves an average reflectance value as low as 0.130%, significantly outperforming both the flock fabric (0.274%) and the textile in the cited study (~0.8%). This comparison, along with a detailed discussion of our process’s simplicity, sustainability, and performance, is now summarized in Table S5 and S6. We believe this choice provides a more rigorous and relevant performance context for our unique fabrication method.

Reviewer #2:

The use of plasma etching is well-documented. However, the study does not adequately demonstrate how this approach significantly improves upon current ultrablack materials in terms of cost, scalability, or end-use performance.

The study provides limited validation for its proposed applications. Although it mentions potential uses in wearable textiles and optical camouflages, more experimental evidence is required to demonstrate UBW’s performance under real-world conditions such as environmental exposure or prolonged use.

Author:

We have now included a cost and process comparison between ultrablack material from carbon nanotubes and textile based ultra black materials in Table S5 and S6. This outlines the cost effectiveness and simplicity of our procedure. In addition to the cost and process benefits, the ultrablack wool is flexible, biocompatible and breathable while showing real world usage by its confirmed mechanical strength, compressibility and wash durability (Section 2.4).

We've also incorporated the reviewer's suggestions by conducting additional experiments. These new findings offer promising directions for future research. Building on our current study, which highlights the ultrablack optical phenomenon in a fabric form factor, we further explored its application in camouflage by examining the fabrics under night-vision and thermal cameras (in Fig. S12 - 16).

Our night-vision images show that UBW appeared the darkest when compared to several commercial fabrics, including the commercially available ultrablack flock material, which we have been referring to as the flock fabric. This is due to polydopamine (PDA)'s intrinsic near-infrared (NIR) absorbing properties. We also considered other NIR-absorbing flock fabrics (sister products to the ultrablack flock) that aren't typically considered ultrablack in the visible spectrum, with total reflectance measured to be around 0.9%. (Fig. S13-15) It is important to note that none of the flock fabrics we tested were truly "wearable fabrics" or "conventional textiles" as they all contained a polymer composite layer as a lamination method, coating layer, or an adhesive back. Such layer prevents air and moisture permeability and eliminates any stretchability or flexibility property of general textiles.

Finally, while it would not be the main interest for this study, thermal camera imaging confirmed that neither the PDA dyeing nor the plasma etching had any negative impact on the fabric's thermal properties. In fact, the ultrablack flock fabric displayed higher thermal emission. (Fig. S16)

Regarding the validation for real-world applications such as environmental exposure or prolonged use, we are currently conducting experiments to maximize the benefits of the ultrablack property and the fabric's intrinsic characteristics in two key areas: enhanced thermoregulation and improved sustainability.

Reviewer #2 Comments end here

Reviewer #3 (Remarks to the Author):

The authors present a simple method for producing wool fabric with low reflectance, including at a wide range of incident angles. I think the simplicity of the approach and the excellent optical performance makes this work significant. In order to facilitate comparison with other work, I think calibration/testing with a low reflectance reference standard should be shown. Also, I think the simulation section is hard to follow and needs to be expanded to make the arguments convincing. Below are several questions that I think should be addressed:

Author:

We thank the reviewer for acknowledging the simplicity and excellent performance of our work. All of reviewer's valuable concerns, questions, and comments have been answered below, and we have modified our manuscript as well as the Supplementary Information document.

Reviewer #3:

1) Can the authors provide photographs with high light intensity and/or longer exposure time to provide some information about the homogeneity?

Author:

We thank the reviewer for highlighting the need for additional photographic comparisons of fabrics treated under different plasma conditions. In response, we have added new representative photographs in Fig. 1 and Fig. S1 to more clearly illustrate the visual differences under various lighting conditions and exposures.

Reviewer #3:

2) Can the authors provide a measurement of a reference standard to provide confidence in the quantitative values of reflectance? I think this is important for repeatability of this work and accurate comparison of quantitative values between different groups.

Author:

We thank the reviewer for raising this important point regarding the need for a reference standard to ensure confidence, repeatability, and consistency in reflectance and transmittance measurements.

Ensuring the accuracy and reproducibility of our data is a top priority. To that end, all measurements were conducted in close collaboration with equipment managers and subject-matter experts, whose protocols and guidance we followed rigorously. Their contributions have been acknowledged in our manuscript.

As described in the Methods section, we used an Agilent Cary 5000 spectrophotometer equipped with an integrating sphere, a setup widely regarded as a gold standard for total reflectance measurements. Unlike directional reflectance techniques, the integrating sphere captures both specular and diffuse components, which is particularly critical for characterizing surfaces modified to enhance light scattering or absorbing.

To ensure reliability across all samples, key parameters such as sample size, aperture size, and beam size were kept constant. The system was calibrated before each measurement session using a certified reflectance standard. The calibration procedure involved two steps: first, the reflectance of the reference standard was measured; second, the beam entrance was blocked to define the 0% transmittance baseline. After this calibration, the reference standard was remeasured to confirm the baseline accuracy.

We have now included a plot of the calibration process below. The average total reflectance across the full measured spectrum ($\lambda = 200\text{-}2500\text{ nm}$) was 99.972%, and across the visible range ($\lambda = 400\text{-}700\text{ nm}$), it was 99.984%, confirming the high precision of our measurements.

Reviewer #3:

3) Is reflectance or transreflectance more important for applications? If just reflectance, couldn't you just make larger gaps to have more light transmit through?

Author:

We appreciate the reviewer for this thoughtful question. We have now clarified the terminologies, significance, and our intent in the manuscript. The goal of our study is to achieve an ultrablack textile by minimizing both reflectance and transmittance, thereby maximizing light absorption. While increasing transmittance (e.g., by introducing larger gaps) may reduce reflectance, it would do so at the expense of allowing more light to pass through the material, which directly undermines the ultrablack effect. Our intent is not to trade reflectance for transmittance, but to suppress both simultaneously through structural and surface modifications that trap incident light.

At the same time, we place strong emphasis on preserving the intrinsic properties of the textile, such as breathability, flexibility, and mechanical integrity. Our fabrication approach avoids coatings or lamination of additives, ensuring that the textile remains suitable for practical, wearable, or large-area applications.

To support our claim of suppressed reflection and transmission, we conducted transreflectance measurements, which serve a dual purpose. First, transreflectance, defined as the total light that is either reflected or transmitted, directly illustrates our goal of minimizing both transmittance and reflectance. This is particularly relevant because many ultrablack surfaces achieve low reflectance by allowing high transmittance or relying on absorbing backing material. In our case, fabric alone functions as both the ultrablack material and the substrate, without additional absorbing layers or rigid supports. Second, transreflectance was measured across a range of incidence angles (-60° to 0 to 60°), offering a more comprehensive evaluation of angular stability of the ultrablack effect. This is essential because some ultrablack surfaces, such as flocked materials, can appear black only at normal incidence but exhibit noticeable reflectance at oblique angles (Supplementary Video and Fig. S11). Our material, by contrast, maintains symmetric transreflectance across varying angles, highlighting its angle-independent performance. In fact, the transreflectance data as well as the visual assessment indicates that UBW absorbs more light in oblique angles than in right angle because of the transmittance from open structure of the fabric.

Along the same lines, we acknowledge the difficulty of directly comparing our textile-based system with engineered ultrablack materials like vertically aligned carbon nanotube arrays (VANTAs). VANTAs are typically grown on rigid, opaque substrates that inherently block transmission, allowing absorbance to be measured directly. In contrast, textiles are semi-permeable by nature and would show higher transmittance unless specifically engineered to suppress the transmittance. This makes absorbance an unsuitable comparison metric. Reflectance and transreflectance, therefore, offer a fairer and more rigorous basis for evaluating our fabric's optical performance, especially given its standalone functionality without the need for separate substrate.

Reviewer #3:

4) For the measurements in Fig. 2f and 2g, is the transreflectance at normal incidence higher simply because of the larger transmission through holes in the fabric? If so, can the authors do this experiment with a thicker patch of wool (or maybe two pieces stacked on top of one another) in order to minimize the transmitted part? I think it is important to isolate the angle-dependent reflectance, because this might be the more relevant thing in an actual application.

Author:

We appreciate the reviewer's important point regarding light transmission through the porous textile, particularly at near-normal incidence. This transmission is a direct result of our design choice to preserve the textile as a single-layer natural fabric without any added coatings, substrates, or composite polymer layers. The observed optical effect stems purely from the engineered nanostructures on the surface of the fabric (Fig. 1 and cross-sectional images from Fig. S2).

While stacking layers or using thicker patches could suppress transmission and isolate reflectance more cleanly, such modifications would depart from the intended real-use configuration. These approaches are analogous to applying laminate or substrate—methods we deliberately avoided to maintain the intrinsic properties of the fabric.

In this context, transreflectance (reflection + transmission) offers a more meaningful and application-relevant performance metric. It captures the overall optical loss from a single textile layer. Despite its open structure, our fabric exhibits transreflectance values comparable to or even lower than flocked fabrics (Fig. 3e), which typically rely on composite layering to eliminate transmission.

Furthermore, stacking fabrics would not only undermine the single-layer design but would also introduce optical complexity. Scattering and absorption between layers make it difficult to attribute the measured signals to the surface effect alone. For instance, doubling fabric thickness does not linearly reduce reflectance or transmission, complicating interpretation. Similarly, the readings may entirely differ when we could, theoretically, remove the substrate from the VANTA structure to measure the absorbance or remove the polymer layer from the flock material to measure the reflectance.

For these reasons, we believe that evaluating the surface-level optical performance of our fabric—without stacking or backing—is both rigorous assessment and directly aligned with our design principles.

Reviewer #3:

5) I don't understand how the authors are incorporating the nanofibrils within the pillars. Can the authors provide some more explanation? Are the nanofibrils the same as the pillars? What is meant by "surface coupling effects at the individual nanofibril interfaces"?

Author:

We thank the reviewer for raising this concern. For simplicity, we have changed the term "pillars" to "bundles of nanofibrils" or in short "bundles" throughout the manuscript.

To answer the reviewer's question, "bundles" are formed from collapsed nanofibrils. We also include a schematic in the manuscript and additional graphical representations in Fig. S4 to avoid this confusion. Having said that, the term "surface coupling effect" was used to refer to the scattering events at the nanofibrillar surface, this term is no longer used in the manuscript. For clarity and to provide comprehensive and more conclusive results, we have modified the section on optical simulations and provided better explanations for the observed optical phenomena.

Reviewer #3:

6) Is what is labeled as D in the SEM in the supplement supposed to be W? And then is D something else? Also, D and d are not labeled in Figure 3.

Author:

Thank you for pointing out this oversight. D and W refer to the diameter of the pores and the width of the bundles (or pillars as per original manuscript), respectively, which we have now correctly marked (Fig. 4a, S4 and Table S1)

Reviewer #3:

7) From the SEMs, the geometry changes significantly across the sample, so I think the authors need to investigate the impact of geometry parameters on the simulation results.

We appreciate the reviewer for raising this concern. We have now included optical simulations capturing the effect of various parameters of the nanofibrils and bundles. The observation highlighted in the original manuscript that the hierarchical structure (which includes nanofibrils, bundles as well as nanopores) is critical for the enhanced darkness remains an accurate conclusion. In addition, we outline the effect of bundles, the density of nanofibrils and dimensions such as H and W on the optical behavior. Furthermore, the simulation model has been refined to more accurately represent the behavior of the actual structures.

Reviewer #3:

8)Page 9, the magnitude of the force in the compression test doesn't seem relevant by itself. I think the authors should convert this to an approximate average pressure and explain how this approximate value is determined.

Author:

We have converted the magnitudes of forces to pressure (in units of kPa and psi) and included average values. Please note that the threshold pressure value for ultrablack behavior reported in the revised manuscript is higher. This change resulted from replacing the square-shaped steel plate in the Instron system with a circular steel plate (diameter: 3.8 cm) matching the fabric's diameter. This modification improves the accuracy and uniformity of the applied pressure by reducing the influence of the anisotropic macroscale structure of the knitted wool fabric. These changes are detailed in the Methods section.

Reviewer #3:

9) The claim in the abstract that the material shows “an omnidirectional ultrablack property, even after compressive loading...” is not substantiated as far as I can tell. The authors do not show angle-dependent integrating sphere measurements after the compressive load test.

We appreciate the reviewers' comments in bringing forth this oversight. We have included the angle -variable integrating sphere measurements as well as the %R, %A and %T measurements for the UBW fabric compressed at 2370 kPa ($L^* \sim 5$). The measurements confirms that the fabric remains ultra black (% R_{avg} of 0.465%). Unlike the original UBW, the spectrum is asymmetric due to compressed nanofibrils slanting towards one direction, giving rise to an angle bias, amidst

this, the fabric retains its UBW effect across the wide range of incident angles (-60° to $+60^\circ$) as observed from Fig. S18.

Minor comments from Reviewer #3:

10) On page 3, the sentence starting with “A coloration meant...” does not seem grammatically correct.

This sentence is now corrected.

11) The authors should probably add labels to the different patches in 2a and 2b

We have added lower case roman numerals.

12) Page 11, missing unit on 0.01

Unit is now included.

13) I think Figure 1 should also show the samples without plasma processing to understand the starting structure

Figure 1 now shows photographic and SEM images of samples without plasma processing.

Reviewer #3 Comments end here

We thank all three reviewers for taking time to review our manuscript and for providing valuable feedback that has greatly improved our work. Please find below our responses to Reviewer 1 and 3 in **purple**. All changes are marked in red in the manuscript by enabling “track changes” option.

Reviewer #1

I appreciate Authors’ effort on the revision. Most of my concerns have been now addressed. Therefore, I recommend the revised manuscript for publication after minor revisions.

1. As the authors noted in their rebuttal, the avian structure is primarily in the microstructure. Then, it would be better to slightly revise the title. Instead of "avian nanostructure," "hierarchical avian structure" or "avian microstructure" might better describe this study.

Author:

We appreciate the reviewer’s valuable input since the first round of revisions, which has significantly improved our work. In response to this comment, we also agree that “Hierarchical Avian Structure” is more appropriate and have revised the manuscript accordingly.

2. Can the author clarify whether the PDA dye is present on the surface or the inner surface of the wool fibers after 2 hours of immersion dyeing? Ref. 30 in the revised manuscript suggests that PDA adheres to the surface of the silk yarn when dyed. Since plasma etching is performed on fabrics pre-dyed with PDA for more than 30 minutes, if the PDA layer thickness or penetration depth is thin, the dyed PDA may be completely etched onto the upper surface of the wool fiber, affecting the blackness.

Author:

During the PDA dyeing process, two pathways occur: (i) monomeric and intermediate units of dopamine penetrate the fiber surface, and (ii) hydrophobic–hydrophilic interactions lead to PDA aggregate formation, which then deposits on the fiber surface. The reviewer is partially correct - the PDA particles gets deposited on the surface, and are etched out during the first 5 minutes of plasma treatment, but this is not a concern for us. Our focus is on the first pathway, namely dye penetration into the fiber (‘chemical absorption’ rather than ‘physical adsorption’), which results in chemical/molecular-level modification of the wool fiber, as evidenced by XPS and FTIR.

The presence of PDA is confirmed from its black color. In our work, PDA dye penetrates across the entire fiber, similar to the findings of Ref. 34 (in manuscript). The dye penetration is confirmed

by cross-sectional images of the fibers, which demonstrate that plasma etching of the outer surface does not remove PDA *inside* the fiber. For comparison, we also provide cross-sectional images of undyed wool and plasma-etched wool fibers in below *Figure a -c*. These show that the color of the fiber cross-section remains black (unchanged) after etching. We also understand that the terminology can be confusing, even in the above-referenced study (Ref. 34) and Ref. 30 mentioned by the reviewer, the authors of the studies use the terms PDA *coating* and *dyeing* interchangeably, although predominantly dyeing has occurred, while the SEMs confirm some-level of coating. In textile terminology, *dyeing* penetrates the fibers embedding the color within the textile, while *coating* applies the material onto the surface of the fabric

In summary, for creating any ultrablack material, both the pigment and structural effects are important. If the PDA were completely etched out, we would not be able to achieve the ultrablack effect. As shown in the control sample below (*Figure 1: d and e*), where a PDA *coating* was applied to the fabric and then plasma treated, the fabric became lighter as the PDA was etched from the surface. This confirms that a traditional PDA *coating* should be avoided, and that PDA penetration into the fiber *or dyeing* is essential for creating ultrablack fabric.

Figure 1: Cross section of **a.** undyed wool, **b.** PDA dyed wool, **c.** Ultrablack wool **d.** PDA-coated textile. **e.** Top view of the same fabric. The small cutout at the top left corner (blue dotted square) shows the fabric after plasma treatment, where the PDA coating has been etched away.

The white specks observed in **d.** are due to the undyed bulk of the material. Thus, an ultrablack textile cannot be achieved by plasma etching a PDA-coated textile.

--End of Response to Reviewer 1--

Reviewer #3

Thank you to the authors for the additional details. There are still a few important points that are critical for articulating the novelty of this work and therefore justifying publication in this journal:

Author:

Thank you for your thoughtful feedback. Answering your comments has greatly helped us to clarify our claims, redefine the terminology, and expand our analysis. We have provided a point-by-point response to your questions below. First, our responses to your comments #1, #2, and #3 are closely related. To comprehensively address them, we have provided chronological explanation and included a new experimental data that supports our claims across all three points. Also, your comments #6, #8, and #9 were closely related. Therefore, in response, we have provided new data that strengthens and clarifies the findings.

1) In the rebuttal letter, the authors claimed that their goal was to not to trade reflection for transmission (i.e. goal was to maximize absorption). Therefore, the transreflectance measurements are most important. However, in terms of TR, the flocked fabric outperforms the UBW at all angles except +50 and +60 degrees (Figure 3e). Can the authors comment on this?

Author:

We thank the reviewer for this very relevant question. Your observation of the transreflectance (%TR) data is correct.

Firstly, the flock fabric's lower %TR (and lower %T) is a direct result of its composite layered construction, as shown in Fig. S9. It uses thick polymer backing that, while effective at holding the vertical filaments and blocking light, means the material sacrifices the properties of a conventional fabric. In direct contrast, our UBW's extreme darkness comes **only** from the nanofibril structures created on the fabric's surface. Because this modification is purely superficial, the underlying knitted fabric is not affected at all. UBW achieves such a high level of optical performance without compromising the fabric's natural properties is a key finding of our work.

Secondly, this structural difference becomes most apparent when viewing the materials at different angles. As we describe on page 11 and 12 of our manuscript and in numerous photographs in the SI and in Fig. 1g and 1h, our UBW maintains a consistent, deep black appearance regardless of the viewing angle. The flock fabric, on the other hand, exhibits random, shiny reflections, which makes it appear less dark under many lighting conditions.

Therefore, our claim is not that our UBW is numerically “darker” on every single measurement. Rather, our main contribution is a practical and simple method that can be scaled up to create material that meets the scientific definition of ultrablack ($\%R < 0.5\%$)¹⁻³ while remaining a completely functional and conventional fabric.

References:

1. McCoy, D. E., Feo, T., Harvey, T. A. & Prum, R. O. Structural absorption by barbule microstructures of super black bird of paradise feathers. *Nature Communications* **9**, 1–8 (2018).
2. McCoy, D. E. *et al.* Structurally assisted super black in colourful peacock spiders. *Proceedings of the Royal Society B: Biological Sciences* **286**, 20190589 (2019).
3. Davis, A. L. *et al.* Ultra-black Camouflage in Deep-Sea Fishes. *Current Biology* **30**, 3470–3476 (2020).

2) Also, if absorption is most important, then wouldn't it make most sense to define “ultrablack” in terms of absorption rather than reflection? If this is done, then the UBW would only be ultrablack from -30 to -60 degree and +30 to +60 degrees, right?

Author:

We agree with the reviewer that for a solid, opaque material, absorption would be the most holistic metric for defining an “ultrablack” state. However, we are not stating that absorption is the most important aspect. Our primary goal was to achieve this specific surface light-trapping property on a textile, not to alter its bulk structure. Therefore, we have referenced and re-stated the term “ultrablack” based on the reflectance metric.

The central focus of our study is the surface modification of the fibers themselves. We intentionally preserved the fabric's open macro- and micro-pores because blocking them would compromise the very properties (like breathability) that make a textile useful. Our aim was to make the fabric ultrablack, not to create an opaque, non-breathable sheet.

As the plots in Fig. 3 show, the transmittance ($\%TR$) of UBW at lower viewing angles (near-normal incidence) is dominated by transmittance, not reflectance. This transmittance is an intended and direct consequence of maintaining the fabric's porous structure. The success of our surface modification, however, is now demonstrated in the new Supplementary Figure S12 (also inserted as part of our response to your next comment), which we conducted in response to your first three comments. By placing a 0 $\%T$ backing material, the result confirms that $\%R$ remains suppressed across all incident angles, while proving that $\%T$ was the primary contributor to the higher $\%TR$ at near-normal incidence.

Therefore, using the established, reflectance-based definition of “ultrablack” is the most accurate way to evaluate the success of our surface treatment, which was the main objective of this work.

3) To be clear, I understand why the authors originally defined “ultrablack” based on reflectance (namely that this is the definition used in other works), but I think this could be misleading because the absence of transmission could be implied in such a definition. This is why I was wondering if the authors could consider characterizing their material with a backing material. I appreciate that the authors didn’t want to “cheat” by putting a backing material, but I think UBW would always be used with some material behind it (please correct me if that is wrong). Would it make sense to characterize with a realistic backing material (perhaps human skin or some other material that might be beneath the UBW)? This would allow the authors to have a realistic metric for the amount of light that would be reflected from UBW in a real-world situation.

Author:

We thank the reviewer for this excellent point. We agree that for transparent and semi-transparent material, relying solely on reflectance (%R) can be ambiguous. As mentioned briefly in the response to your previous comment, your suggestion prompted us to conduct a new experiment to isolate the surface reflectance (%R) from the transmittance (%T), providing a much clearer picture of our material’s ultrablack performance.

To eliminate the influence of %T, we placed our UBW fabric directly onto a backing material with zero transmission: 3M Scotch black electrical tape (T ~0%, R ~5%). This setup (UBW-0%T backing material) allows us to measure the light reflected from the UBW’s nanofibril surface without any contribution from light passing through the fabric.

%R, %T, and %TR data for this new setup are presented here and are added to SI as Figure S12.

As above plots demonstrate, removing the transmitted component dramatically lowers the measured %TR signal at near-normal angles. %R plot clearly demonstrates the performance of UBW by showing near identical behavior with or without the backing material. Along with %TR plot, the new curve effectively represents the true reflectance from the UBW's surface, combined with any light that is internally scattered within the fabric before being absorbed. This result directly confirms the highly efficient light-trapping nature of our surface modification, independent of the fabric's inherent porosity.

4) “it shows a low reflectance at all probed angles”...I believe it should say “transflectance”.

Author:

Thank you for the careful inspection. We have now made the correction accordingly.

5) There seems to be substantial nonuniformity in UBW, which is particularly noticeable when illuminated at off-normal angles (Fig 1h). Can the authors comment on the cause of the nonuniformity? How significant are the changes in lightness across the sample? Also, will this nonuniformity be significant in envisioned applications? Is this non-uniformity more significant after compression?

Author:

Thank you for your careful inspection of the figures. We would like to offer a clarification regarding Figure 1g and 1h. The image labeled “iv” is flock fabric and “v” is our UBW fabric. We believe the substantial non-uniformity you’ve noted is a characteristic of the flock fabric’s surface, as seen in Figure 1h. In contrast, our UBW demonstrates a highly consistent and uniform dark appearance. This uniformity is further supported by the additional photographs in the SI and the Supplementary Video.

As described in our earlier responses, the combination of visual assessment, SEM analysis, and optical measurements provides a more comprehensive and intuitive understanding of our material’s properties.

7) I do not think the statement of “omnidirectional ultrablack” is substantiated. In the discussion of Figure 3, the authors point out that the material is only confirmed to be ultrablack (defined by authors as $R < 0.5\%$) near normal (from integrating sphere reflectance) and between -60 to -30 degrees and $+30$ - $+60$ degrees (from angle-dependent TR). This should be clarified in the abstract so as not to overstate the accomplishment. If the authors want to claim ultrablack performance across the whole range, they will need to decouple T and R somehow. However, given that the authors indicate that TR is the important metric, I am now not sure if isolating R is the right thing to do.

Author:

We thank the reviewer for this important point, and we agree. The term “omnidirectional ultrablack” may be an overstatement, as our measurements do not cover a full 180° range. We acknowledge your valid concern that this could be confusing for readers.

Based on your valuable suggestion, we have replaced this term with “wide-angle ultrablack.”

We believe this term is well-supported by our data, which demonstrates that reflectance remains below 0.5% across a wide 120° angular span (from -60° to +60°). Furthermore, with the newly added Figure S12, we indeed observe ultrablack property across the wide incident angles. To ensure full clarity for the reader, we have now explicitly defined this term by this performance metric in the revised text.

6) After the compression test, UBW no longer outperforms flocked fabric. If the authors think the mechanical robustness is a significant point, then they should compare compressed flocked fabric to compressed UBW.

Author:

We agree with your assessment. We have conducted new %TR measurements on both compressed materials (now added as Figure S20 in the SI) and have also tested the UBW with a 0%T backing material (as newly added in your previous comment as Figure S12 in the SI). As we have responded to your first comment, our focus is not on outperforming the flock fabric in every measurement. We use the %TR and visual analysis data, however, to demonstrate the significance of our findings, highlighting similarities and differences when contrasted with the ultrablack flock fabric. As you can see, the effects of compression on the two fabrics are distinctly different.

8) From Fig. S18d, after compression, the material is only confirmed to be ultrablack ($R < 0.5\%$) at normal incidence because all TR values presented for the compressed material are above 0.5%. The abstract and text should be revised accordingly.

Author:

Based on this new data, we have revised the abstract and the manuscript. We have retracted the claim that our material remains ultrablack across wide incident angle “even after compression”. Instead, we now highlight the UBW’s ability to suppress the severe optical bias that flock fabric exhibits post-compression, a finding that parallels data from the super black bird-of-paradise study¹.

References:

1. McCoy, D. E., Feo, T., Harvey, T. A. & Prum, R. O. Structural absorption by barbule microstructures of super black bird of paradise feathers. *Nature Communications* **9**, 1–8 (2018).

9) It seems like the flocked fabric has the issue of fibrils not being straight, but the wool seems to have a similar issue (specifically Figure S18d is asymmetric) after mechanical loading. Is it possible that the flocked fabric has seen load already during packaging, shipping, and handling and that is why it isn’t good at off-normal angles? From experience with flocked fabric, different regions can exhibit different fibril alignment which can be modified by mechanical load. I think this is an important consideration because only the pristine (i.e. not compressed) UBW shows the superlative properties.

Author:

We are confident that the newly added data resolves your concern regarding sample handling. By subjecting both materials to identical mechanical pressure, we have demonstrated that UBW’s superior performance is an intrinsic property. This performance gap is a direct result of their fundamental structural differences. As shown in the supplementary SEM images (Figure S9), the flock’s tall micro-filaments (1~1.5 mm) would readily bend and collapse under pressure, creating a strong directional reflectance bias. In contrast, UBW’s hierarchical nanofibril structure resists this deformation allowing it to maintain consistent, symmetric, wide-angle optical performance even after significant mechanical loading.

--End of Response to Reviewer 3--